


# Seasonal analysis of submicron aerosol in Old Delhi using high resolution aerosol mass spectrometry: Chemical characterisation, source apportionment and new marker identification

James M. Cash[1,2], Ben Langford[1], Chiara Di Marco[1], Neil Mullinger[1], James Allan[3], Ernesto Reyes-Villegas[3], Ruthambara Joshi[3], Mathew R. Heal[2],W. Joe F. Acton[4,Δ], Nick Hewitt[4], Pawel K. Misztal[1,*], Will Drysdale[5], Tuhin K. Mandal[6], Shivani[7], Ranu Gadi[7] and Eiko Nemitz[1]

[1] UK Centre for Ecology & Hydrology, Edinburgh Research Station, Penicuik, EH26 0QB, UK
[2] School of Chemistry, University of Edinburgh, Edinburgh, EH9 3FJ, Edinburgh, UK
[3] Department of Earth and Environmental Sciences, University of Manchester, Manchester, M13 9PL, UK
[4] Lancaster Environment Centre, Lancaster University, Lancaster, LA1 4YQ, UK
[5] Wolfson Atmospheric Chemistry Laboratory, University of York, York, YO10 5DD, UK
[6] CSIR National Physics Laboratory,
[7] Department of Applied Sciences and Humanities, Indira Gandhi Delhi Technical University for Women, Delhi, 110006, India

[Δ] Now at: School of Geography, Earth and Environmental Sciences, University of Birmingham, Birmingham, UK
[*] Now at: Department of Civil, Architectural and Environmental Engineering, The University of Texas at Austin,
Austin, TX 78712, USA

*Correspondence to*: James Cash (jacash94@ceh.ac.uk) and Eiko Nemitz (en@ceh.ac.uk)

**Abstract**

We present the first real-time composition of submicron particulate matter ($PM_1$) in Old Delhi using high
resolution aerosol mass spectrometry (HR-AMS). Old Delhi is one of the most polluted locations in the world, and $PM_1$ concentrations reached ~600 µg m$^{-3}$ during the most polluted period, the post-monsoon, where $PM_1$ increased by 178% over the pre-monsoon period. Using positive matrix factorisation (PMF) to perform source apportionment analysis, two burning-related factors contribute the most (35%) to the post-monsoon increase. The first PMF factor, semi-volatility biomass burning organic aerosol (SVBBOA), shows a high correlation with earth
observation fire counts in surrounding states which links its origin to crop residue burning. The second is a solid-fuel OA (SFOA) factor with links to local open burning due to its high composition of polyaromatic hydrocarbons (PAH) and novel AMS measured marker species for polychlorinated dibenzodioxins (PCDDs) and polychlorinated dibenzofurans (PCDFs). Two traffic factors were resolved, one hydrocarbon-like OA (HOA) factor and another nitrogen-rich HOA (NHOA) factor. The N compounds within NHOA were mainly nitrile
species which have not previously been identified within AMS measurements. Their PAH composition suggests that NHOA is linked to diesel, and HOA to compressed natural gas and gasoline. These factors combined make the largest relative contribution to primary $PM_1$ mass during the pre-monsoon and monsoon periods, while contributing the second highest in the post-monsoon. A cooking OA (COA) factor shows strong links to the secondary factor, semi-volatility oxygenated OA (SVOOA). Correlations with co-located volatile organic
compound (VOC) measurements and AMS measured organic nitrogen oxides (OrgNO) suggest SVOOA is formed from aged COA. It is also found that a significant increase in chloride concentrations (488%) from pre-





monsoon to post-monsoon correlates well with SVBBOA and SFOA suggesting that crop residue burning and open waste burning are responsible. A reduction in traffic emissions would effectively reduce concentrations across most of the year. In order to reduce the post-monsoon peak, sources such as funeral pyres, solid waste burning and crop residue burning should be considered when developing new air quality policy.

## 1. Introduction

The number of premature deaths linked to fine particulate matter ($PM_{2.5}$) globally was estimated at 4.1 million in 2016 and 10.6% were in India alone (Gakidou et al., 2017). Some of the most polluted cities in the world are in India, with Delhi being in the top 10 cities listed by the World Health Organisation based on available annual $PM_{2.5}$ measurements in 2018 (World Health Organisation, 2018). This has a significant effect on the health of its population (26 million) and in 2016 it was estimated that fine particulate matter ($PM_{2.5}$) accounted for ~15,000 premature deaths per year (Maji et al., 2018). Therefore, it is important to investigate the sources of PM to improve air quality mitigation strategies.

The composition of sub-micron PM ($PM_1$) is a mixture of organic aerosol (OA), black carbon (BC) and inorganic aerosols including ammonium, sulphate, chloride and nitrate. The high-resolution aerosol mass spectrometer (HR-AMS) has helped to improve measurements of $PM_1$ as it gives detailed information on its chemical composition through elemental analysis and high-resolution compound identification. Combining these measurements with positive matrix factorisation (PMF), allows for the apportionment of aerosol mass into different sources (or factors). This provides detailed information needed to inform effective air quality interventions. To date, relatively few measurements of PM composition have been made by HR-AMS in India with measurements made in Kanpur a notable exception (Chakraborty et al., 2015, 2016a, 2016b, 2018). These HR-AMS measurements however give multiple factor solutions, often of six or greater, which highlights the complexity of the source mix. Typical sources identified include components considered to be primary such as hydrocarbon-like organic aerosol (HOA) and biomass burning organic aerosol (BBOA) as well as components usually considered to be secondary such as low-volatility oxygenated organic aerosol (LVOOA), semi-volatility oxygenated organic aerosol (SVOOA), and oxidised-BBOA (O-BBOA). Throughout these studies there are multiple versions of the same source and requires additional auxiliary measurements to pin down their origin. This study aims to better assign factors to their original sources using additional measurements of volatile organic compounds (VOCs), black carbon (BC), carbon monoxide (CO), nitrogen oxides ($NO_x$) and Earth observations.

A growing number of studies in Delhi, and other locations in India, have reported large concentrations of chloride, especially during the morning hours at ~7-9 a.m. (Sudheer et al., 2014; Chakraborty et al., 2018; Gani et al., 2019; Acharja et al., 2020; Reyes-Villegas et al., 2020). The source of this chloride is still widely debated and there are several possibilities including aluminium pickling, industrial activity, municipal waste burning and biomass burning. The aluminium pickling industry in the north west of Delhi is thought to directly emit HCl and be the main contributor to chloride during the winter months when average concentrations increase from 1.5 µg m$^{-3}$ in the summer to 23 µg m$^{-3}$ in the winter (Gani et al., 2019). Studies in Mexico city suggest a substantial amount of chloride is from biomass and municipal waste burning and they also observe high concentrations from crop residue burning (Christian et al., 2010; Li et al., 2012). In Delhi, these sources may also be high contributors as it would explain why chloride concentrations show a larger elevation during the colder post-monsoon and winter months than other compounds.





During the post-monsoon, large-scale crop residue burning occurs along with large numbers of Delhi residents burning municipal waste to keep warm and clean streets. These open burning sources produce chloride in different organic and inorganic forms. Inorganic $NH_4Cl$ is related to biomass burning through the release of particulate KCl that forms HCl and ultimately $NH_4Cl$ once reacted with ammonia (Sullivan et al., 2007; Wang et al., 2017). Emissions of HCl are often associated with plastic burning but there are also organic species of chloride, such as polychlorinated dibenzo-furans (PCDF) and -dioxins (PCDD), that form readily from chloride-rich plastics such as polyvinyl chloride (PVC) and polychlorinated biphenyls (PCB) (Minh et al., 2003; Chakraborty et al., 2013; Stewart et al., 2020). Large-scale plastic burning practises in Delhi, such as electronic waste recycling centres situated in the North West, could therefore be high contributors to chloride mass. The measurement of plastic burning markers is therefore important to understand the contribution of such practises to chloride concentrations. This study focuses on the high-resolution aspect of the High-Resolution Time-of-Flight Aerosol Mass Spectrometer (HR-TOF-AMS) and investigates the detailed composition of aerosol. Using Positive Matrix Factorisation (PMF), the $PM_1$ source mass profiles in Delhi are compared to auxiliary measurements, including high-resolution proton transfer reaction mass spectrometer (PTR-QiTOF) measurements of VOCs, giving us a detailed picture of the origin of PMF factors. The polyaromatic hydrocarbon (PAH) composition of each factor is also investigated and helps to distinguish factors based on the combustion of different fuel types. A focus is also placed on investigating the origin of the high chloride concentrations observed in Delhi, where compositional aids help to establish a mixture of chloride sources and an indication to their contribution. There are two companion papers which use the HR-TOF-AMS measurement data from IGDTUW. One study compares AMS datasets measured at different sites in Delhi and creates a city-wide perspective of $PM_1$ composition over an entire year (Reyes-Villegas et al., 2020). The other study (which will be published separately) uses the unit-mass resolution measurements to calculate compositional aerosol fluxes for the first time and compares measurements in Delhi to measurements taken in two cities: London and Beijing. The three studies combine to produce a more comprehensive picture of $PM_1$ aerosol composition, concentrations and sources in Delhi.

## 2. Methods

### 2.1. Measurement location and instrumentation

Measurements were undertaken on the campus of the Indira Gandhi Delhi Technical University for Women (IGDTUW) which is located to the north of Old Delhi (28°39'51.8"N 77°13'55.2"E). The measurements were made in three campaigns corresponding to three periods of the year in Delhi where the seasonal differences cause significant changes in meteorology (Figure 1). Here, we refer to the measurements taken during the summer months as the pre-monsoon period (26/05/2018 – 28/06/2018) when average temperatures were high (36 °C) and relative humidity (RH) was low (43%). During the monsoon campaign (03/08/2018 – 18/08/2018) the temperatures dropped slightly (31 °C) and RH was high (76%) coinciding with large rainfall events. The post-monsoon campaign (09/10/2018 – 23/11/2018) encompassed the periods of the 5-days of Diwali (05/11/18 – 10/11/18) and the seasonal stubble burning for rice crops in the neighbouring states of Punjab and Haryana situated to the NW. These two activities are combined with unfavourable meteorology where boundary layer height, temperatures (24 °C) and wind speeds are generally lower, causing a shallow inversion layer.



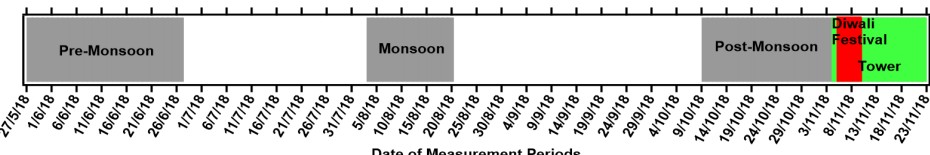

**Figure 1 – Gant chart showing the measurement periods during the pre-monsoon, monsoon and the first half of post-monsoon periods. The red region shows the Diwali festival period and the green region shows when the inlet was moved to a 30 m tower.**

A High-Resolution Time-of-Flight Aerosol Mass Spectrometer (HR-TOF-AMS, Aerodyne Research Inc.) was used in V-mode to measure 5-minute average mass spectra of non-refractory $PM_1$. The principle of operation of the instrument is described in detail elsewhere (DeCarlo et al., 2006; Canagaratna et al., 2007). Briefly, particles are sampled via a ~$PM_1$ aerodynamic lens and their size is measured using a time-of-flight chamber. Particles are then vaporised on a heated plate (600 °C), the non-refractory species ionised using a 70 eV electron impact source and the *m/z* values of the resultants ions determined with a time-of-flight analyser.

The HR-TOF-AMS was calibrated throughout the campaigns for its ionisation efficiency of nitrate (IE) and the relative ionisation efficiency (RIE) of other inorganic compounds using nebulised 300 nm ammonium nitrate, sulphate and chloride. A collection efficiency (CE) of 0.5 was confirmed through comparisons with $PM_{2.5}$ filter measurements taken throughout the pre- and post-monsoon campaigns. The comparison gave $PM_1$ vs. $PM_{2.5}$ gradients of 0.9 using a CE of 0.5 during most of the measurement campaigns. Considering the increased mass collected due to a higher size fraction for $PM_{2.5}$, the CE was set to 0.5 for these periods. However, for Diwali and a few days after (05 to 14/11/18) a higher CE of 1 was applied as a more accurate $PM_1$ vs. $PM_{2.5}$ gradient of ~0.8 was found when set to CE=1. The reasons for this change are not the purpose of this study and further compositional analysis is required to understand the change.

The HR-ToF-AMS data were analysed using the IGOR Pro (Wavemetrics, Inc., Portland, OR, USA) based software SQUIRREL (Sequential Igor data Retrieval) v.1.62B and PIKA (Peak Integration by Key Analysis) v.1.22B and a total of 2725 ions were fitted (including isotopes) in the range of *m/z* 12-328. This range extends to larger *m/z* than is typically used for ambient measurements and certainly for high resolution PMF calculations using AMS data (Aiken et al., 2009; Docherty et al., 2011; Sun et al., 2016; Zhang et al., 2018). This becomes possible due to the high concentration loading in Delhi (up to ~600 µg m$^{-3}$) providing considerable signal at high *m/z*. The fitting of heavier ions has also been shown to improve the strength of PMF solutions and their correlations with external tracers, as demonstrated in the Supplementary Information (SI) Section S1. However, fragment identification becomes increasingly ambiguous at higher *m/z.* Therefore, only peaks which significantly improved the open and closed signal residuals were fitted regardless of the residuals in the difference (diff = open – closed) signal. Peaks were also fitted based on signal intensity where peaks lower than 2 Hz/ns were not included.

The extended *m/z* range included ions identified as polyaromatic hydrocarbons (PAH) which were fitted according to the protocol set out by Herring et al. (2015). Elemental analysis was also performed to derive oxygen-to-carbon (O:C), hydrogen-to-carbon (H:C), nitrogen-to-carbon (N:C) and organic mass to organic carbon (OM:OC) ratios using the Improved-Ambient (IA) method developed by Canagaratna et al. (2015) as implemented in PIKA.

The concentration of organic nitrogen oxide species (OrgNO) have been estimated using the $NO_2^+/NO^+$ ratio ($R_{obs}$) method described in detail by Farmer et al. (2010). The method involves calculating the fraction of OrgNO ($OrgNO_{frac}$) using the equation,





$$OrgNO_{frac} = \frac{(R_{obs} - R_{cal})(1 + R_{ON})}{(R_{ON} - R_{cal})(1 + R_{obs})}. \tag{1}$$

where $R_{cal}$ is the $NO_2^+/NO^+$ ratio observed during $NH_4NO_3$ calibrations and $R_{ON}$ is a fixed value (0.1) that has been determined as the lowest $R_{obs}$ in multiple data sets (Kiendler-Scharr et al., 2016). The calculated $OrgNO_{frac}$ is then multiplied by the total nitrate concentration to give an estimate of the OrgNO mass ($OrgNO_{mass}$).

The ISORROPIA-II model was used to estimate the liquid water content (LWC) in aerosol using the input parameter of temperature and relative humidity along with AMS measurements of ammonium, sulphate, chloride and nitrate (Fountoukis and Nenes, 2007).

The AMS measurements were compared to a number of co-located instruments including black carbon (BC) measurements which were taken using an Aethalometer AE-31 and corrected for by a Single Particle Soot

Photometer (SP-2; Droplet Measurement Technology, Boulder, CO) (Reyes-Villegas et al., 2020). Volatile organic compounds (VOC) were measured using a high-resolution proton transfer reaction mass spectrometer (PTR-QiTOF, Ionicon Analytik G.m.b.H., Innsbruck, Austria). A 5-minute average of the concentrations were used within this work and details of the setup are described in (Acton et al., 2020). CO and $NO_x$ concentrations were measured using an Aerolaser AL 5002 UVU and a dual-channel high-resolution chemiluminescence (Air

Quality Designs Inc., Colorado), respectively. Meteorological measurements were taken using a HS-50 Gill research ultrasonic anemometer and a Vaisala weather transmitter (WXT530 Series) from the 11/10/2018 onwards. All measurements before this date were taken from the Indira Gandhi International Airport (available at: https://www.ncdc.noaa.gov/cdo-web/).

**2.2. Source apportionment**

Positive Matrix Factorisation (PMF) is a bilinear receptor model which is used as a multivariate analysis tool to separate the total measured mass spectrum and its corresponding time series into individual sources or "factors" (Paatero and Tapper, 1994). The model can be summarised as:

$$X = FG + E, \tag{2}$$

where $X$ is the total measured matrix containing $m$ rows of mass spectra for each time stamp and $n$ columns of measured $m/z$ ions. The factor mass spectral profiles are contained within the $m \times p$ matrix $F$ and the time series contribution of each factor is expressed by the $p \times n$ matrix $G$. The residuals of the model are represented by the $m \times n$ matrix $E$ and the number of factors, $p$, are chosen by the model user but it requires no a priori information for matrices $F$ and $G$. The model iteratively fits $F$ and $G$ to the data in order to minimise the fitting parameter, $Q$,

through a least-squares algorithm as:

$$Q = \sum_{i=1}^{m} \sum_{j=1}^{n} \left(\frac{e_{ij}}{\sigma_{ij}}\right)^2. \tag{3}$$

Here the elements of the residual, $e_{ij}$, and uncertainty, $\sigma_{ij}$, matrices have rows, $i$, and columns, $j$, where in ideal circumstances $Q = 1$ and the model would create a minimised solution that would explain all the measured data and leave the uncertainty within the residual matrix.

The analysis was carried out using the PMF Evaluation Tool (PET, v3.00) using the PMF2 algorithm in robust-mode (Ulbrich et al., 2009). The error matrix was down-weighted based on the signal-to-noise ratio (SNR) where columns with an SNR <2 were down-weighted by a factor of two and SNR <0.2 by a factor of 10. Down-weighting





of $CO_2^+$ and its associated (derived) ions $CO^+$, $H^+$, $OH^-$, $H_2O$, was also included to account for the influence of over-counting.

The primary PMF analysis was conducted on the organic matrix and a 7-factor solution was chosen following the procedure and rationale set out in SI Sections S1 and S2. The procedure also includes multilinear regression analysis to help determine the chosen solution. The solution chosen is from PMF analysis carried out on all the measurement periods combined as multilinear regressions showed this to fit more closely with external tracers (see Section S1 for more details).

An alternative PMF solution was obtained, introducing inorganic species into the data matrix using the method set out by Sun et al. (2012). Briefly, selected ions for chloride ($Cl^+$ $m/z$ 35, $HCl^+$ $m/z$ 36), ammonium ($NH^+$ $m/z$ 15, $NH_2^+$ $m/z$ 16, $NH_3^+$ $m/z$ 17), nitrate ($NO^+$ $m/z$ 30, $NO_2^+$ $m/z$ 46) and sulphate ($SO^+$ $m/z$ 48, $SO_2^+$ $m/z$ 64, $SO_3^+$ $m/z$ 80, $HSO_3^+$ $m/z$ 81, $H_2SO_4^+$ $m/z$ 98) were chosen and it was confirmed that once combined to form one time series they correlated almost perfectly with the measured total mass of their represented inorganic species ($r^2$ ~1).

All adjustments for relative ionisation and collection efficiencies were applied to the data before PMF analysis in order to determine accurate uncertainties and allow reasonable mixing of the organic and inorganic data. After PMF analysis, the selected ions were then divided by the total mass fraction of their represented inorganic species. This time a 9-factor solution was chosen (Section S3) and was developed partly to provide an analysis product that would be comparable with the earlier Aerosol Chemical Speciation Monitor (ACSM) PMF results for Delhi

of Bhandari et al. (2019). Additionally, it provides information on the association between the organic factors and the ions that are normally associated with inorganic compounds.

Although there is some ambiguity in the choice of the ideal solutions, PMF was chosen over the more advanced Multi Linear engine (ME2; Canonaco et al., 2013), which allows individual factors to be constrained with a priori information because, for India, reference spectra have not yet been established and especially not for high

resolution spectra covering the wide range of $m/z$ used in our analysis.

## 3. Results

### 3.1. Inorganic and organic PM concentrations

The concentrations of PM species during the pre-monsoon and monsoon periods are similar although slightly

lower in the latter, presumably due to increased rainfall and associated washout (Figure 2). However, most PM component concentrations were considerably higher during the post-monsoon period and total-$PM_1$ mass increases by 178% over the pre-monsoon. Chloride and nitrate concentrations increase the most, by a factor of ~5-6, suggesting there is a significant additional source of chloride and nitrate during the post-monsoon period. This is also consistent with increased partitioning of (semi-)volatile aerosol compounds into the aerosol phase

during the cooler post-monsoon period. Ammonium and organic aerosol underwent a ~2-3 fold increase, whereas sulphate is the only species whose absolute concentrations did not increase during the post-monsoon. This has also been noted in previous measurements taken in Delhi using an ACSM where sulphate was on average 10 μg $m^{-3}$ across spring, summer and the monsoon period (Gani et al., 2019).



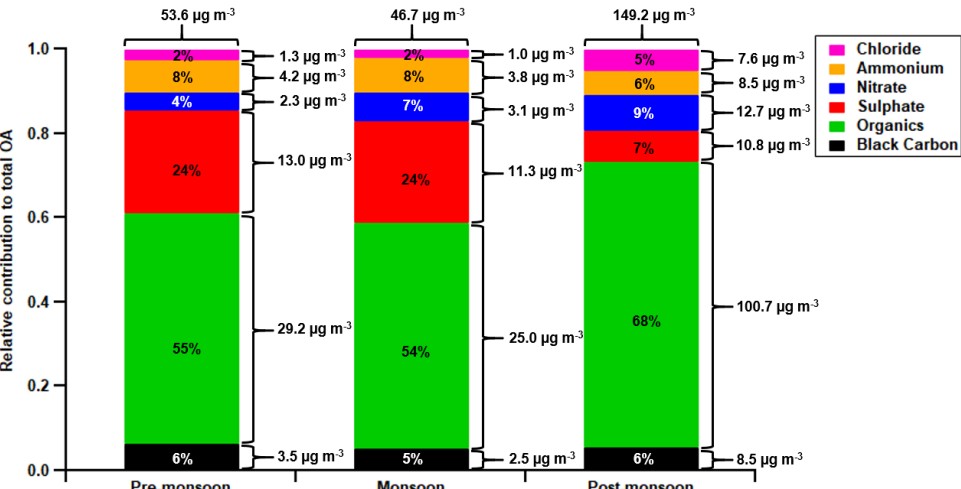


**Figure 2 – Average relative contributions of chloride, ammonium, nitrate, sulphate, organic aerosol and black carbon to the total PM$_1$ mass loadings in the pre-monsoon, monsoon and post-monsoon periods. The average concentrations of each species are shown to the right of each bar.**

Preliminary analyses of the eddy-covariance flux measurements (Ben Langford pers. commun.) made during the post-monsoon tower period suggest that whilst concentrations peaked at night, local emissions peaked during the late morning and afternoon, suggesting that boundary-layer dynamics exercised a dominant control over surface concentrations. Concentrations of all species, with the possible exception of sulphate, decreased significantly in the afternoon during the post-monsoon (Figure 3). This was mainly due to the contraction of the boundary layer

during the night and early hours of the morning being more pronounced which removes possible afternoon peaks from specific sources, e.g. lunch hour cooking activities (Nakoudi et al., 2019; Reyes-Villegas et al., 2020). The pre-monsoon and monsoon diurnal cycles show species concentrations, excluding nitrate and chloride, were less affected by the higher boundary layer in the afternoon. The monsoon is the least affected period as there were midday peaks for sulphate, ammonium and organic aerosol.


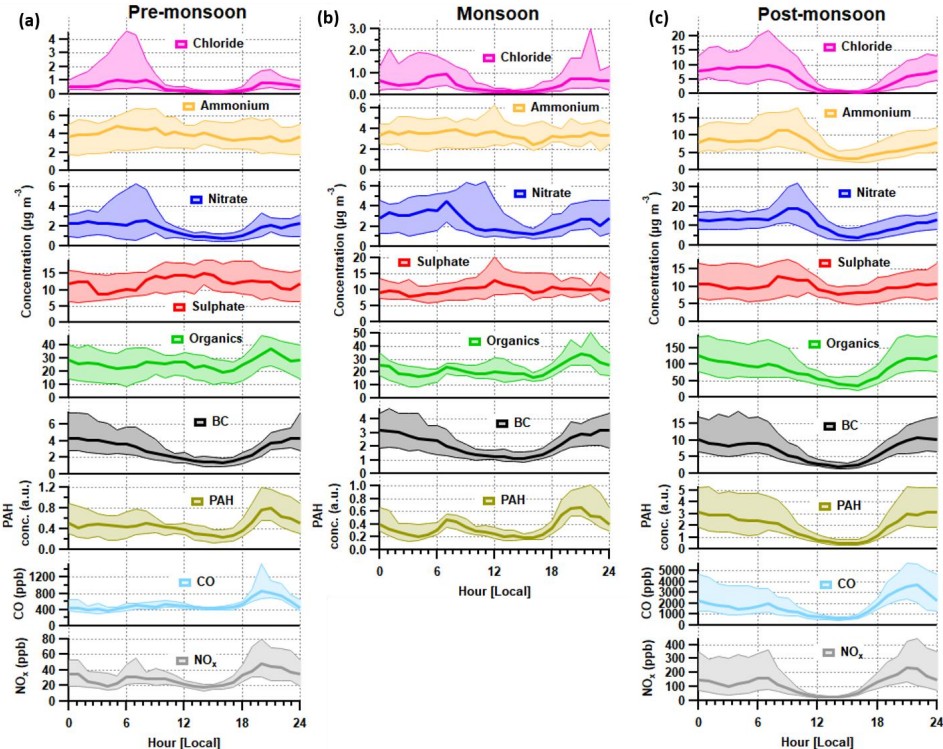

**Figure 3 – Median diurnal cycles for aerosol chemical species and for BC, CO and NO_x during the (a) pre-monsoon, (b) monsoon and (c) post-monsoon periods. The median concentration is represented by the thick line and the interquartile range is represented by the shading. Data for CO and NO_x are not available for the monsoon period.**


Morning peaks were observed in the diurnal cycle for nitrate, ammonium and chloride (Figure 3). The nitrate peaks are earlier in the morning during the pre-monsoon compared to other periods which may relate to an earlier rise in temperature compared to monsoon and post-monsoon periods. The diurnal cycle for chloride shows a regular peak at ~ 6-8 a.m. and the details of possible sources will be discussed in Section 4.3.1.

The pollution roses (showing the concentration frequency distribution by wind sector, Figure 4) for nitrate and ammonium show higher contributions to $PM_1$ mass with a south-westerly wind direction but there is no clear pattern. Figure S8 shows polar graphs (i.e. the wind-sector distribution of the raw 30-minute measurements) which in particular illustrate the wind direction corresponding to maximum concentrations. Nitrate and ammonium show scattered polar graphs which indicates no clear source directions were associated with extreme concentrations.

The sulphate pollution rose shows a slightly higher mass contribution with a south easterly wind but its polar graph shows maximum concentrations originated from the north and north east.

The pollution rose and polar graph for organic aerosol is highly spread. The pollution rose suggests that most of the organic mass is from the east and south east with high peaks (> 140 µg m⁻³) originating from the west and northwest. Its polar graph also shows some extreme values existing from the south east. This spread can be

explained by a low wind speed and a low boundary layer height causing a significant increase in concentrations. In Figure 5, $PM_1$ shows a large increase during periods of low wind speed and an exponential decay in $PM_1$





concentration with increasing wind speed. Meteorology therefore plays an important role during the high concentration periods.

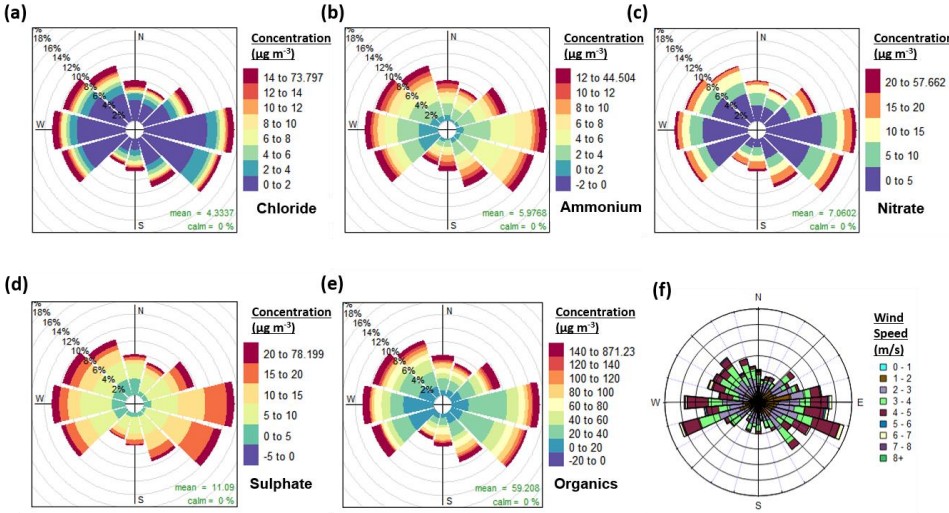

**Figure 4 – Pollution roses for (a) chloride, (b) ammonium, (c) nitrate, (d) sulphate and (e) organic aerosol, along with (f) a wind rose plot for all measurement periods combined. The pollution roses show the frequency of counts as a percentage for each 30° wind vector. The counts are divided into concentration bins based on the colour scale in the legend.**


The chloride pollution rose (Figure 4a) shows no clear pattern in its wind direction and the polar graph in Figure S8 is spread, showing large chloride peaks which coincide with the south east, east, northeast and north. The pollution roses for each diurnal hour in Figure S9 show the directional trend of chloride in more detail. Here, the large peak in the morning (6-8 a.m.) corresponds to a mixed spread of wind directions with a preference for the west, south west, southeast and northwest which suggests mixed sources of chloride.







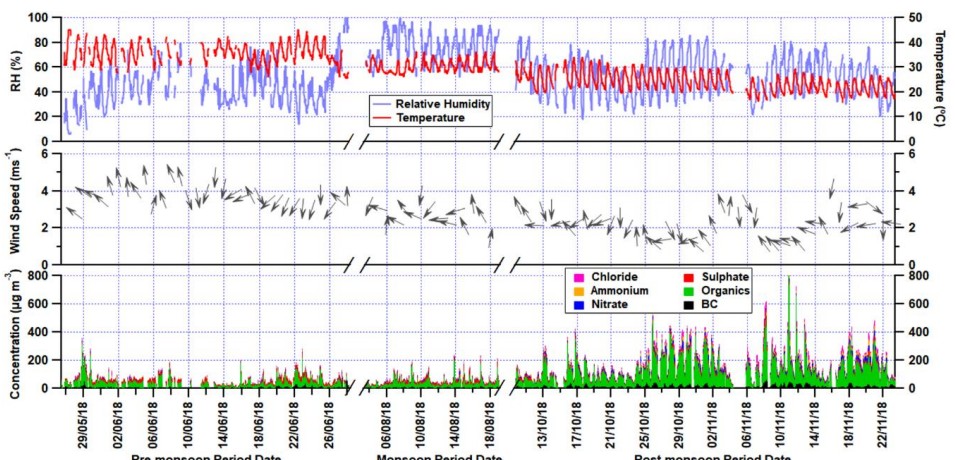

**Figure 5 – Upper panel: time series of the relative humidity and the temperature for the three measurement periods. Middle panel: time series of the wind speed with arrows showing wind direction. Bottom panel: time series of stacked concentrations of aerosol species showing total PM$_1$.**


### 3.2. Concentrations of polyaromatic hydrocarbons and OrgNO

The presence of OrgNO species is evident in Figure 6 and there is a gradual drop in concentration with time over the pre-monsoon, monsoon and earlier half of the post-monsoon period. This may suggest an increase in temperature and a decrease in relative humidity are key to OrgNO formation in Delhi. During Diwali (07/11/2018)

there is also a sharp increase in OrgNO concentrations to > 20 µg m$^{-3}$ and this is likely due to large amounts of pyrotechnical emissions. Similar peaks in AMS-measured OrgNO have been seen during a large pyrotechnical event in the UK called bonfire night (Reyes-Villegas et al., 2018b).

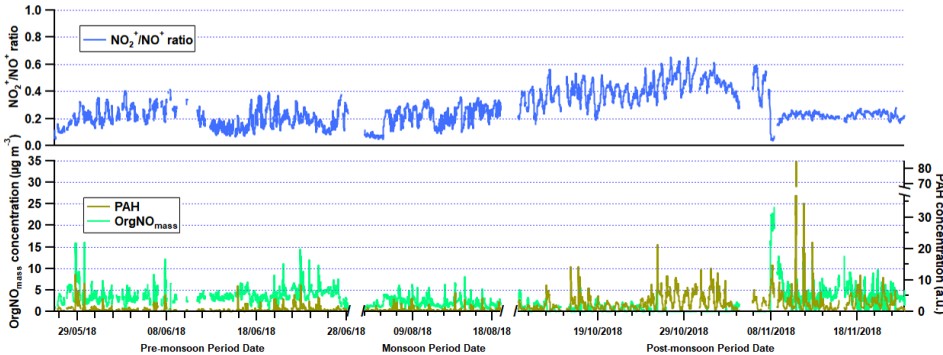

**Figure 6 – Upper panel: time series of NO$_2$$^+$/NO$^+$ ratio in the three measurement periods. Lower panel: time series of polyaromatic hydrocarbon (PAH) uncalibrated concentrations and organic nitrogen oxide species (OrgNO$_{mass}$) concentrations.**





The PAH time series in Figure 6 shows similar quantities during the pre-monsoon and monsoon periods and the

consistent low levels suggest common sources such as traffic, solid fuel burning or cooking activities. There is however a large increase in PAHs during the post-monsoon period when the burning of the rice crop residue begins, but there are also three large peaks during the week of the Diwali holiday. Each peak occurs during the late hours of the night (~10-12 p.m.) which suggests an infrequent local source. It is uncertain what this source is and through source apportionment analysis (see Section 3.3 below) the PAH contribution to these peaks is split

between sources, suggesting they cannot be resolved through PMF without the unknown source occurring more often than these three occasions.

### 3.3. Source apportionment concentrations

The organic-only PMF solution is presented in Figure 7 and Figure 8 while the organic-inorganic solution is

presented in the SI (Section S8). The organic-inorganic solution is not considered the most representative of sources, as it may be influenced by thermodynamic effects, i.e. factors may be resolved based on similarities in volatility rather than source. However, the organic-inorganic solution is referred to within this section as it provides insight into the combined behaviour of inorganic and organic species.

The 7-factor organic solution includes two hydrocarbon-like organic aerosol (HOA) factors relating to traffic,

where one contains more nitrogen (NHOA) and is characterised by a larger contribution from higher $m/z$ ions including PAHs (see below). Such N-rich HOA factors are rarely reported in the literature, but it was a persistent feature in our PMF sensitivity studies (see SI Section S3). There are also two burning related factors separated mainly by their oxygenation and source origin, which we interpret to reflect primary solid fuel organic aerosol (SFOA) and a semi-volatility biomass burning organic aerosol (SVBBOA), also on the basis of the correlation

with regional fire counts (see Section 4.3 below). A cooking organic aerosol (COA) factor is also resolved along with two oxygenated organic aerosol factors (OOA) which are separated based on their volatility: low-volatility OOA (LVOOA) and semi-volatility OOA (SVOOA). Using multilinear regressions, the traffic and burning factors were determined to better correlate with external tracers when separated (see SI Section S3). This is further supported by a PMF study on PTR-MS measurements taken in Delhi which also resolved two traffic and two

burning factors (Wang et al., 2020). A summary of the Pearson's $r$ correlation coefficients for the combined and separated factors are shown in Table S1.



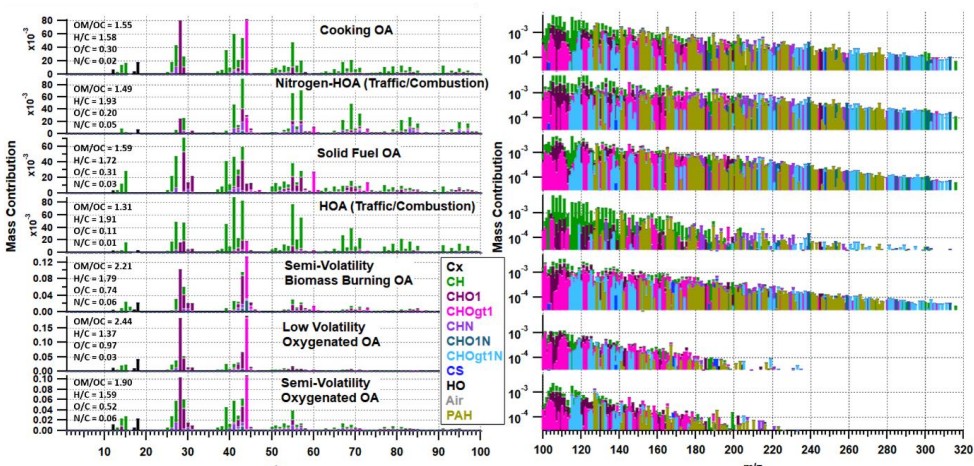

**Figure 7 – Organic-only PMF solution with elemental ratios shown for each factor in the left-hand corner of each spectrum. The mass spectra on the left show *m/z* 12-100 on a linear scale, while the spectra to the right show *m/z* 100-320 on a logarithmic scale. The peaks are coloured based on the chemical families shown in the legend.**

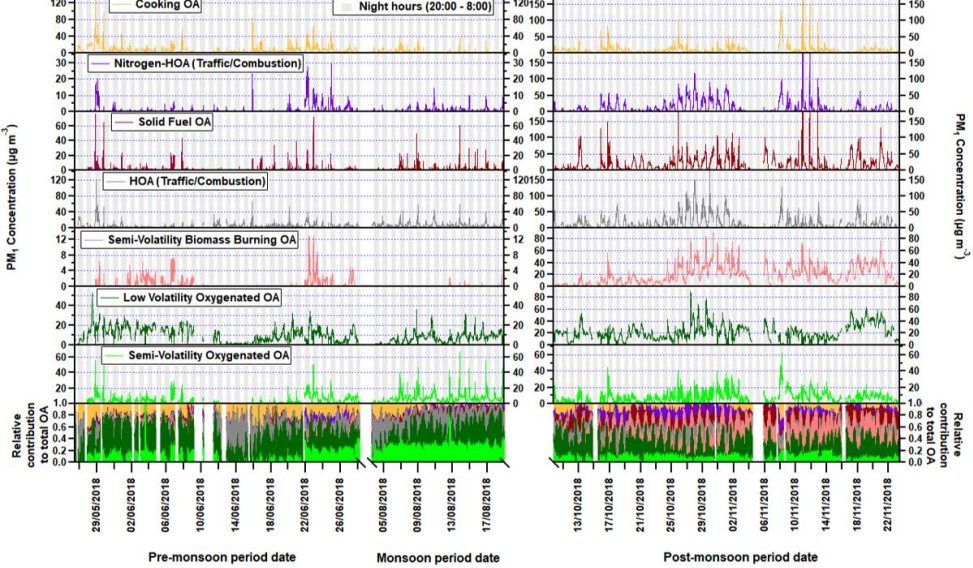

**Figure 8 - Time series for each factor where the x-axis is broken to show each measurement period. Regions shaded in grey are night hours and the time series of the normalised concentrations are also shown in the bottom section of the graph.**

### 3.3.1. Hydrocarbon-like organic aerosol (HOA)

Both HOA and NHOA have characteristically large peaks at 55 and 57 *m/z* along with little evidence of oxidation (Figure 7). Their H:C values are the highest amongst the identified factors and they have the strongest correlations with measurements of BC, CO and $NO_x$ (Figure 9). The diurnal cycle for both factors shows a clear peak in the



evening when traffic is busy. However, the 9 a.m. morning rush hour is less pronounced, particularly for HOA (Figure 10), which could be due to the lifting of restrictions for commercial and heavy goods vehicles at 11 a.m. causing the morning rush hour peak to carry on into the late morning. The PMF results from the AMS unit-mass resolution flux measurements (Ben Langford pers. commun.) show two traffic factors which peak at 9 a.m. and

11 a.m., which supports this. The two factors are therefore assigned as traffic sources which is supported by strong correlations with co-located measurements of benzene, toluene and ethyl benzene, which are commonly used as tracers for traffic emissions (Slowik et al., 2010; Crippa et al., 2013). There is also a high degree of correlation between acetonitrile and both NHOA and HOA, but the reasons for this are unclear. This may suggest that sources associated with the burning of organic matter are closely related to traffic sources.

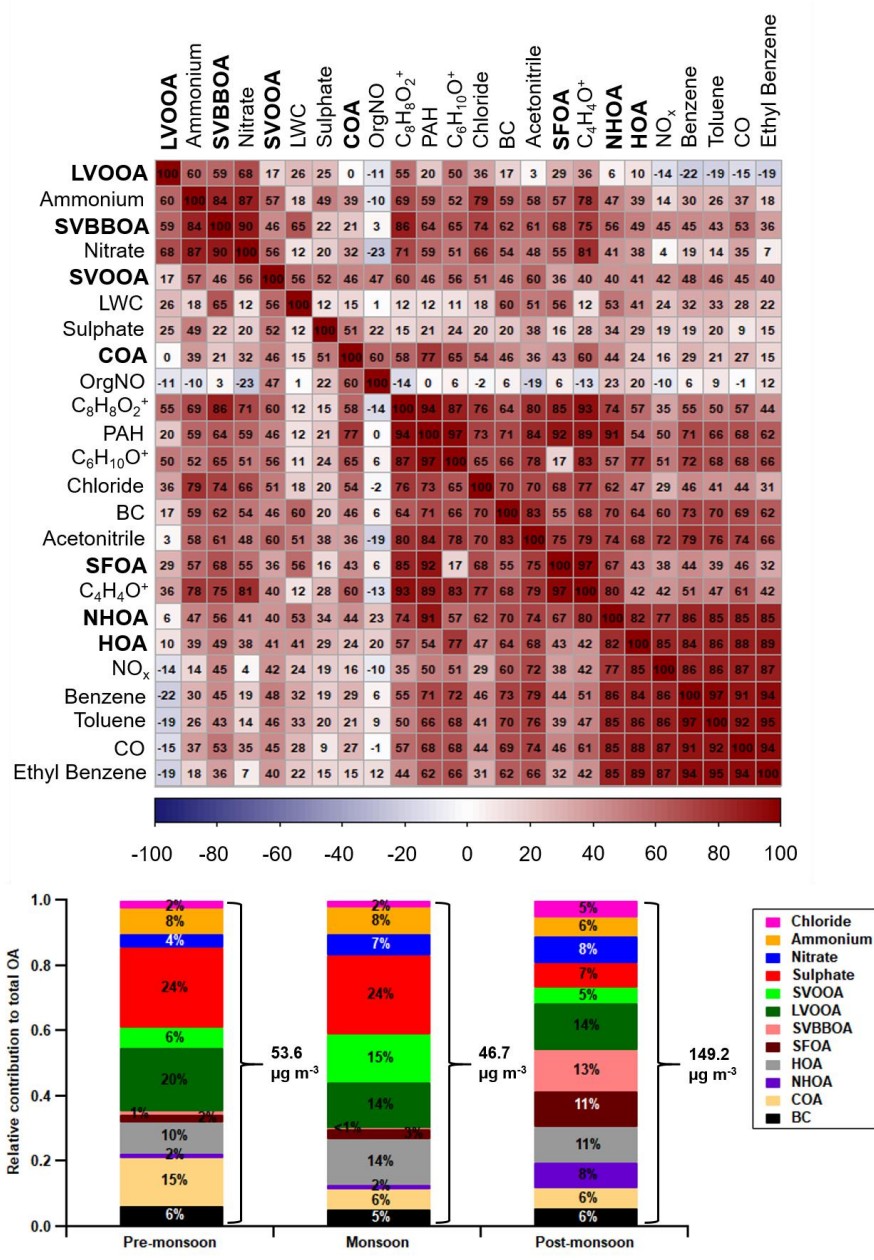

**Figure 9 – Upper panel: Correlation and matrix between the AMS OA factors (bold), internal tracers and external tracers for the combined dataset (all seasons). The correlation coefficients are ordered using hierarchical cluster analysis. Lower panel: relative contribution of OA factors, BC and inorganic species to the total OA for each period where the total average PM$_1$ is shown using right curly brackets (see Table S2 for values and statistics). The three ions [C$_6$H$_{10}$O]$^+$, [C$_8$H$_8$O$_2$]$^+$ and [C$_4$H$_4$O]$^+$ are, respectively: an organic acid fragment used for COA determination, a fragment of dibenzodioxin called benzodioxan and, furan a fragment of dibenzofuran. The abbreviations of VOC species are: Ace = acetonitrile, Ben = benzene, Tol = Toluene and EBen = Ethyl Benzene.**







From Figure 11, a large fraction of HOA mass follows a north westerly wind which points towards a busy
motorway intersection ~0.7 km from the measurement site. There is also a preference for a south westerly wind
which coincides with a nearby train station (~0.5 km) where diesel trains are commonly in use. NHOA shows a
similar preference for both the northwest and southwest however, there is a much larger spread of mass between
wind directions. This may suggest atmospheric processing is occurring, for example oxidation reactions.

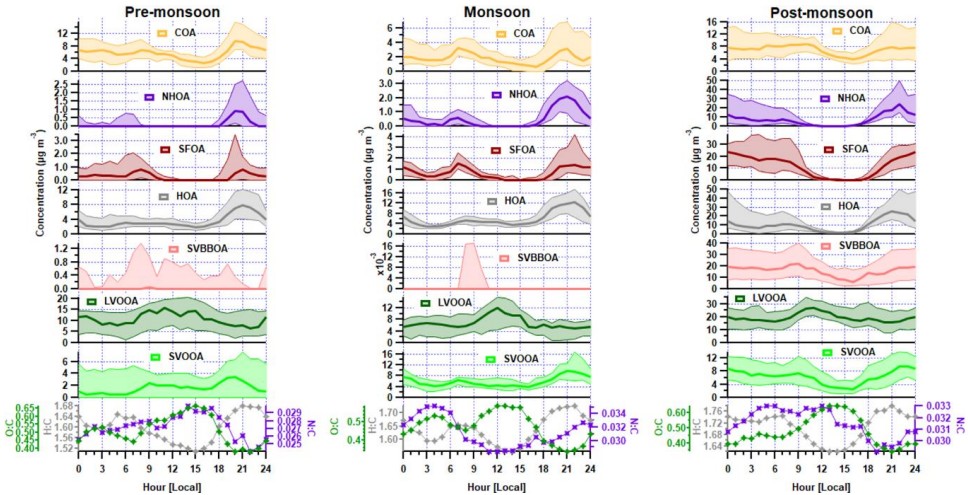


**Figure 10 – Median diurnal cycles of the factor solutions for the three measurement periods (interquartile range
indicated by the shading) along with the elemental ratios.**

The main compositional difference between NHOA and HOA is a higher nitrogen content in NHOA which is
evident in the mass profiles (Figure 7). Peaks at $m/z$ 41 ([$C_2H_3N$]$^+$), $m/z$ 43 ([$C_2H_5N$]$^+$), $m/z$ 55 ([$C_3H_5N$]$^+$), $m/z$ 57
([$C_3H_7N$]$^+$), $m/z$ 83 ([$C_5H_9N$]$^+$), $m/z$ 97 ([$C_6H_{11}N$]$^+$), are all prominent in NHOA but not in HOA. In an attempt to
assess if these peaks are solely responsible for the NHOA factor being resolved, these peaks were systematically
down-weighted during separate PMF analyses by a factor of 2, 10 and 100. The results gave a similar solution
that still separated into two traffic related factors which differed in their nitrogen content due to the above peak
list.

The origin of these ions could be from nitrile compounds (R-C≡N) as the peaks at $m/z$ 41 ([$C_2H_3N$]$^+$) and $m/z$ 97
([$C_6H_{11}N$]$^+$) have shown to be characteristically large for nitriles in standard 70 eV impact ionisation (McLafferty,
1962). They also have an even-to-odd carbon-to-nitrogen preference due to the nitrile fragmentation pattern in
mass spectra (Simoneit et al., 2003; Abas et al., 2004). This is further supported by the comparison of NHOA with
nitrile 70 eV spectra available on the NIST Chemistry WebBook (https://webbook.nist.gov/chemistry) (Figure
S11). The relatively short carbon-chain nitrile compounds, for example, dodecanenitrile, tetradecanenitrile,
hexadecanenitrile and octadecanenitrile, have particularly similar spectra and peak ratios to that of NHOA.



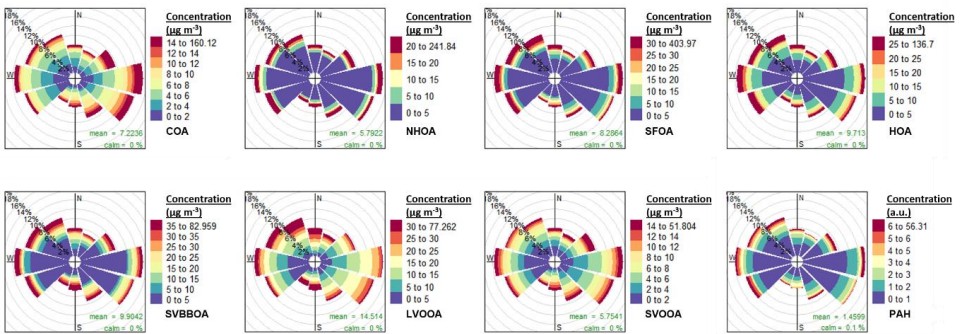

**Figure 11 - Pollution roses for each factor and uncalibrated PAH concentrations. The pollution roses show the frequency of counts as a percentage for each 30º wind vector. The counts are divided into concentration bins based on the colour scale in the legend.**

The PAH content of NHOA and HOA is very different, with the fraction of PAHs within NHOA being high at 3.17 % compared with HOA at 1.00 %. The level of oxidation in the PAH fraction of each factor is shown in Figure 12, where NHOA has the second highest relative concentrations of oxidised PAH (OPAH), after SFOA. A large percentage of OPAH mass is due to the species dibenzodioxin (*m/z* 183 and 184), dibenzopyran and acenaphthoquinone (*m/z* 181 and 182). The presence of dibenzodioxin is also supported by a significant correlation with its mass spectral fragment, benzodioxan ($[C_8H_8O_2]^+$) (Pearson's $r = 0.80$) (Figure 9). NHOA has other high OPAH species including naphthaldehyde (*m/z* 155 and 156) and anthrone (*m/z* 193 and 194) but these are shared amongst other factors as well.


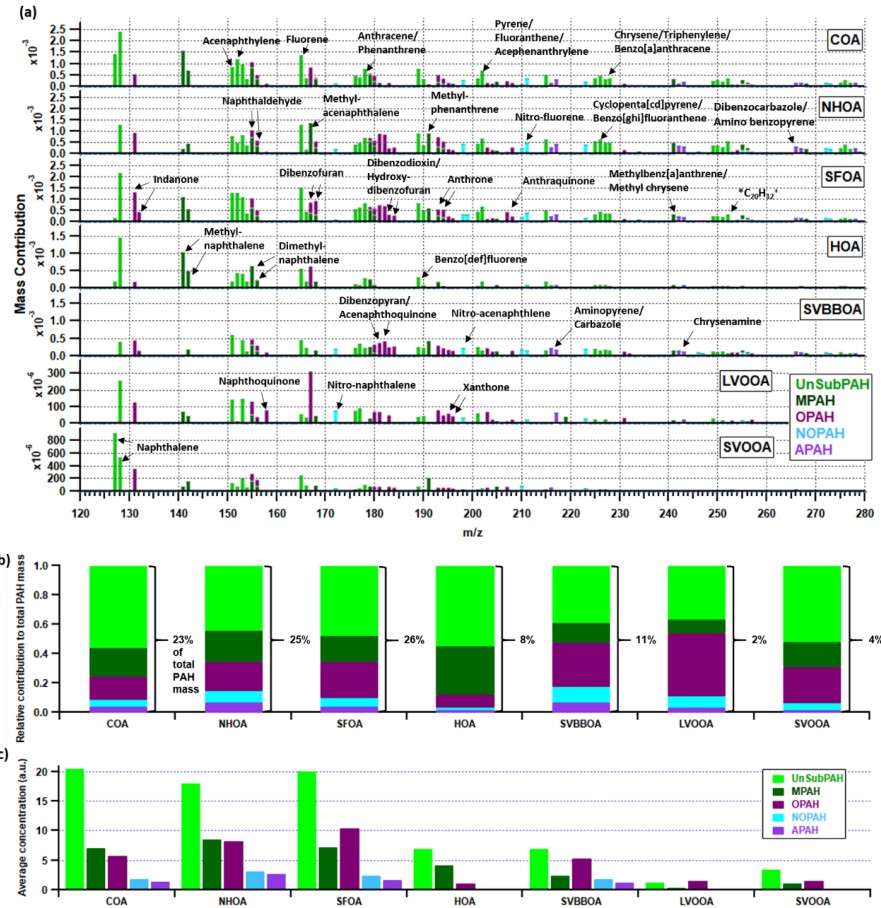

**Figure 12 – PAH factor profiles showing (a) the mass spectra of PAH peaks, (b) the relative contribution to total PAH mass and (c) the average uncalibrated concentrations. The PAH families: UnSubPAH, MPAH, OPAH, NOPAH and**
**APAH, are described within the text and in Herring et al. (2015). \*The peak at *m/z* 252 relating to the ion [C$_{20}$H$_{12}$]$^+$ is a list of PAHs overlapping in mass and includes benzo[b]-, benzo[j]- and benzo[k]flouranthene, along with benzo[a]- and benzo[e]pyrene.**

The NHOA factor contributes the highest relative amounts of amino PAHs (APAH) and nitrogen-oxygen
substituted PAHs (NOPAH) which coincides with this factor containing more nitrogen (Figure 12). The more prominent NOPAH and APAH peaks are nitro-acephthlene (*m/z* 198), nitro-fluorene (*m/z* 210 and 211), aminopyrene and carbazole (*m/z* 216 and 217), nitro-anthracene and nitro-phenanthrene (*m/z* 223), chrysenamine (*m/z* 242 and 243), and dibenzocarbazole and amino benzopyrene (*m/z* 266 and 267). There are also large methyl-substituted PAHs (MPAH) and UnSubPAH peaks in the NHOA factor including methyl-acenaphthalene (*m/z*
167), methyl-phenanthrene (*m/z* 191) and the ion [C$_{20}$H$_{12}$]$^+$ (*m/z* 251 and 252) which represents a number of benzopyrene and benzofluoranthene species that have the same mass (see Figure 12 for full list).

The total mass of traffic-related factors (HOA + NHOA) is the largest of the primary sources (compared with burning = SFOA + SVBBOA and cooking = COA) during the pre-monsoon and monsoon periods. It is also the



joint second largest, similar to SOA (LVOOA + SVOOA), in the post-monsoon period where its average

concentration reaches 27.2 µg m$^{-3}$ (Figure 9).

### 3.3.2. Cooking organic aerosol (COA)

The COA mass profile in Figure 7 has a characteristically high *m/z* 55:57 nominal mass ratio (2.87) and high *m/z* 41 and 43 as seen in previous studies (Allan et al., 2010; Mohr et al., 2012; Sun et al., 2016). The COA time series

also has a moderate correlation with the AMS measured $[C_6H_{10}O]^+$ peak (Pearson's *r* = 0.65) which is often used as an internal tracer for cooking related factors (Sun et al., 2016). The mass spectrum has an uncharacteristically large peak at *m/z* 44 (and the inferred *m/z* 28) and as a consequence a large O:C ratio (0.33) when compared to literature values presented in Table S2. There is also a deviation from literature when observing the Van Krevelen (VK) diagram in Figure S13 as the high O:C pulls it away from a gradient of -2 and towards a gradient of -1. This

relates to a difference in composition from the normal aldehydes and ketones to carboxylic acids. The reasons for this are discussed in Section 4.3.

The diurnal cycle for COA has a clear morning peak across the three periods but it is hard to interpret the midday dynamics because of the pronounced boundary layer affect, whereas in other studies there is a clear lunchtime peak (Figure 10). There is a slight rise in the median at 12 p.m. for the pre-monsoon and monsoon periods but the

post-monsoon period shows a more defined lunchtime peak and supports this factor being related to cooking activities.

The wind rose for COA (Figure 11) is one of the few factors to have a clear pattern and shows that the majority of its mass is coming from the north-east, east and southeast. This is likely from food kiosks, shelters and vendors situated outside the grounds of IGDTUW. The polar graph shows a spread of large peaks matching a northerly

wind (Figure S10) which coincides with the location of shared student kitchens at IGDTUW. The same observations were made by a study measuring n-alkanoic acids at IGDTUW (Gadi et al., 2019). The polar graph also shows maximum concentrations coincide with a south-easterly direction. The timing of these peaks match with maxima seen in polar graphs for PAH, SFOA and NHOA, which suggests a common source.

Figure 13 shows the correlations between ions from the AMS and the seven PMF factors. The AMS correlation

mass spectra offer an alternate view to factor mass spectra. They show which AMS ions correlate strongly with factor time series while removing the influence of ion concentration. This means high *m/z* peaks exist due to their strong association in time and space with a factor, irrespective of the ion signal. A PTR-QiTOF correlation mass spectrum is also shown for the measurements of VOCs taken at the same site during the post monsoon period (Figure 13). This shows COA has very little correlation with hydrocarbon VOCs as the strongest correlations are

with $C_xH_yN_z$, $C_xH_yO_z$ and $C_xH_yN_zO_t$ species. COA also has regions of strong correlations with $C_xH_yN_zO_t$ ions in the later part of both the AMS (*m/z* 210-300) and PTR-MS (*m/z* 120-210) mass spectra (Figure 13). This is consistent with COA having the strongest correlation with OrgNO (Pearson's *r* = 0.60) out of all factors (Figure 9).

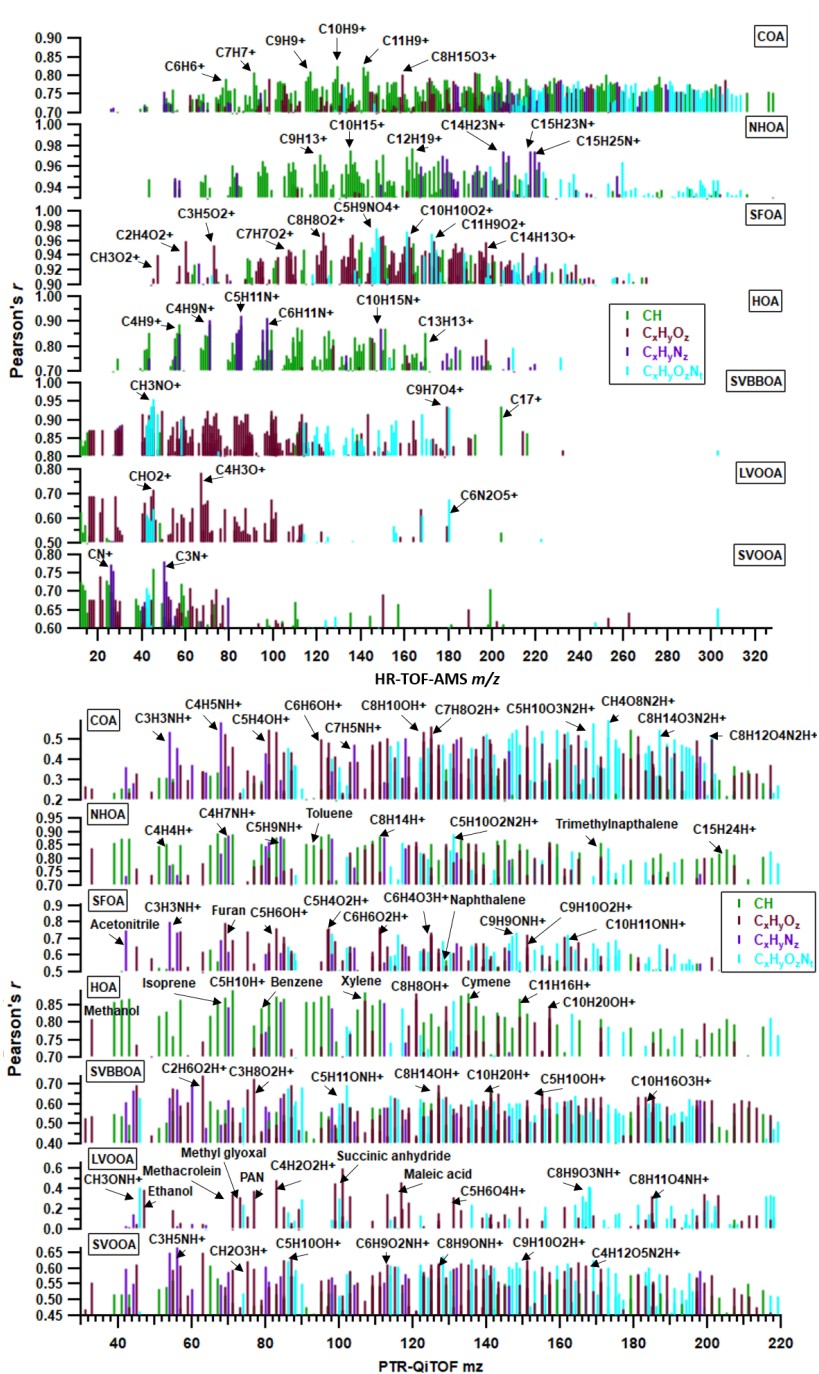

**Figure 13 – AMS and PTR-QiTOF correlation mass spectra where the y-axis shows the Pearson's *r* correlation coefficient between each *m/z* and the respective factor. Peaks are coloured based on the chemical composition described in the legend.**





The COA factor accounts for 23% of the total PAH mass and has a particularly high peak at $m/z$ 91 which is seen in the majority of studies with COA measurements but is not often highlighted (Mohr et al., 2012; Dall'Osto et al., 2013). The fragment ion responsible is $[C_7H_7]^+$ which is seen when measuring PAHs (McLafferty and Turecek, 1993) and is one of five AMS measured PAH fragment ions ($[C_6H_6]^+$, $[C_9H_9]^+$, $[C_{10}H_9]^+$ and $[C_{11}H_9]^+$) that give some of the highest correlations with COA (Figure 13). This is consistent with the PAH composition of COA which is mainly UnSubPAHs and also contributes the largest amount of UnSubPAHs out of all factors (Figure 12). Its defining peaks correspond to naphthalene ($m/z$ 127 and 128), methyl-naphthalene ($m/z$ 141 and 142), acenaphthylene ($m/z$ 151 and 152), fluorene ($m/z$ 165 and 166), anthracene/phenanthrene ($m/z$ 177 and 178), benzo[def]fluorene ($m/z$ 189 and 190), and pyrene/fluoranthene/acephenanthrylene ($m/z$ 201 and 202). These peaks are similar to species seen in literature linked to cooking activities, although it is shown that they largely depend on the fuel used such as, dung cake, wood, etc. (Singh et al., 2010, 2016; Masih et al., 2012; Shivani et al., 2019).

The largest percentage mass contribution of COA to total OA was during the pre-monsoon period (15%) which equates to an average concentration of 3.93 µg m$^{-3}$. There was a relatively small (40%) increase to 5.50 µg m$^{-3}$ in the post-monsoon, the lowest relative increase out of all factors and aerosol species (Figure 2 and Figure 9). This suggests cooking activities were relatively consistent throughout the year. The monsoon period is, however, inconsistent with this interpretation as there was a large 64% decrease in COA which is the largest percentage drop of all factors and aerosol species. This may be linked to meteorology or a decrease in outside cooking emissions as is discussed later.

### 3.3.3. Solid fuel organic aerosol and semi-volatility biomass burning organic aerosol (SFOA and SVBBOA)

The SFOA factor has a strong correlation with gas-phase acetonitrile (Pearson's $r = 0.75$) and has large peaks at $m/z$ 60 and 73 which are the typical AMS biomass burning tracers (Weimer et al., 2008). SVBBOA shares the same peaks at $m/z$ 60 and 73; however, it has a weaker correlation with acetonitrile (Pearson's $r = 0.61$). This may be due to the level of oxidation within SVBBOA as its mass spectrum has a large peak at $m/z$ 44 (and consequently $m/z$ 28), showing evidence that the aerosol is aged.

The O:C ratio for SVBBOA is relatively high (O:C = 0.74) and fits between the ratios for LVOOA and SVOOA (Figure 7) suggesting it is secondary in nature. This is consistent with the VK diagram in Figure S13 which shows SVBBOA lies between the oxidation states ($\overline{OS}_C$) of -1 and 0, typically where LVOOA and SVOOA literature values overlap. Additionally, the SVBBOA appears to be closer to the alcohol functional group gradient ($m = 0$) which may explain the high correlation with LWC (Pearson's $r = 0.65$) due to alcohol functionalised compounds being more water-soluble.

The lower volatility of SVBBOA is also evident in its post-monsoon diurnal cycle as it is less affected in the afternoon by the temperature and boundary layer changes that appear to affect primary emitting factors significantly (Figure 10). SFOA however has more defined peaks in the morning and evening, which is consistent with its aliphatic mass spectral composition, indicating it is fresh primary organic aerosol.

SVBBOA has a particularly high N:C value (0.06) and has prominent $C_xH_yN_zO$ peaks at $m/z$ 43 ($[CHNO]^+$), $m/z$ 44 ($[CH_2NO]^+$) and $m/z$ 45 ($[CH_3NO]^+$), and lower intensity peaks at $m/z$ 42 ($[CNO]^+$), $m/z$ 46 ($[CH_4NO]^+$) and $m/z$ 59 ($[C_2H_5NO]^+$) (Figure S11). This ratio of peaks is characteristic of primary amides where a large contribution from $m/z$ 44 is a result of an α-cleavage fragmentation leaving a $[O=C=NH_2]^+$ ion (Pavia et al., 2014;





Nicolescu, 2017; Fokoue et al., 2018). The absence of a high intensity $m/z$ 59 peak could point to the majority
being low molecular weight amides, because once above three carbon atoms, the amide will undergo a McLafferty
rearrangement and produce a $[H_2N-(C=O)-OH]^+$ ion. It is, however, uncertain how much of an influence the large
$[CO_2]^+$ signal has on the fit of the $[CH_2NO]^+$ peak. The residuals of the peak fitting for the open and closed mass
spectrum improved once $[CH_2NO]^+$ was fitted but this cannot rule out the large overlapping signals from $[CO_2]^+$.
The pollution rose for SFOA has a large spread of low concentration values but for concentrations $> 10$ µg m$^{-3}$
there is a slight dependence to the northwest and east (Figure 11). There are two cremation sites located east and
south east along the Yamuna river (~500 – 750 m) which likely contribute to SFOA mass (Figure S14).
Conversely, high concentrations of SVBBOA occur from the north and south but the frequency is low. This could
suggest its concentration is affected significantly by a drop in wind speed as the north and south winds are slow
and infrequent. Meteorology may, therefore, be a strong influence on the high SVBBOA concentrations.
SFOA and SVBBOA have the strongest correlations with chloride (Pearson's $r$ = 0.68 and 0.74, respectively)
which suggests they have an affinity with chloride sources (Figure 9). The application of the second PMF analysis
where inorganic AMS measurements were combined with the organic matrix (SI Section S8) resulted in a biomass
burning factor which included the majority of the ammonium chloride mass (Figure S15). Additionally, SVBBOA
has the strongest correlations with ammonium and nitrate, out of all factors (Pearson's $r$ = 0.84 and 0.90,
respectively). These results suggest that the high levels of ammonium nitrate, and particularly ammonium
chloride, in the post-monsoon are associated with SVBBOA.
SFOA and SVBBOA have substantially different correlations with PAHs (Pearson's $r$ = 0.92 and 0.64,
respectively) In fact, SFOA has the greatest contribution from PAHs (3.3 %) and contributes 26% to the total
PAH mass (Figure 12). A large portion of its PAH mass is OPAH and this is reflected by its mass spectrum having
defining OPAH peaks such as indanone ($m/z$ 131 and 132), napthaldehyde ($m/z$ 155 and 156), dibenzofuran ($m/z$
167 and 168), dibenzopyran and acenaphthoquinone ($m/z$ 181 and 182), dibenzodioxin and hydroxy-dibenzofuran
($m/z$ 183 and 184), anthrone ($m/z$ 193 and 194), and anthraquinone ($m/z$ 207 and 208). SVBBOA has a similar set
of OPAH peaks but lower in abundance and it lacks dibenzofuran. They both also have prominent UnSubPAHs
such as acenaphthylene ($m/z$ 151 and 152), fluorene ($m/z$ 165) and pyrene/fluoranthene/acephenanthrylene ($m/z$
201 and 202). However, SFOA has a larger naphthalene ($m/z$ 127 and 128) content. This is also seen in the PTR-
QiTOF correlation spectrum where SFOA has a significantly stronger correlation with VOC furan and
naphthalene when compared to SVBBOA (Figure 13).
One of the strongest correlations of AMS ions with SFOA (Figure 13) is the dibenzodioxin fragment, benzodioxan
$[C_8H_8O_2]^+$, and this coincides with high correlations with furan ($[C_4H_4O]^+$) and chloride (Figure 9) which indicates
SFOA may have a strong association with PCDDs and PCDFs.
The two factors SFOA and SVBBOA represent the overall burning-related aerosol. During the pre-monsoon and
monsoon periods they are low in concentration and contribute ~3% to the total organic aerosol mass (Figure 9).
During the post-monsoon, however, they combine to contribute 23% to the organic aerosol mass. The largest
contributor of the two being SVBBOA at ~12% of total PM$_1$ mass, equivalent to ~16.6 µg m$^{-3}$. This increase
means burning-related aerosol is the largest single contributor to the observed organic aerosol mass during the
post-monsoon.





### 3.3.4. Semi-volatility oxygenated organic aerosol and low-volatility oxygenated organic aerosol (SVOOA and LVOOA)

LVOOA and SVOOA are separated based on their volatility and oxygenation and serve as proxies for aged and
less aged secondary organic aerosol (SOA) (Jimenez et al., 2009). The large peak at *m/z* 44 in LVOOA (Figure 7)
implies a high level of oxygenation which coincides with the highest O:C ratio (0.97) out of the resolved factors.
It is also the only factor that consistently peaks in the afternoon, defying the pronounced boundary layer effect,
demonstrating that it is low in volatility and forms through photochemistry (Figure 10). This is further supported
by overall poor correlations with the PTR-QiTOF mass spectrum, where its strongest correlations are mostly with
$C_xH_yO_x$ family VOC species, such as methyl glyoxal and methacrolein (Figure 13b). SVOOA has slightly stronger
correlations with VOCs but the majority are still comparatively poor (Pearson's $r < 0.6$). The strongest correlations
are with ions from the $C_xH_yN_x$ and $C_xH_yN_xO_t$ families which coincides with SVOOA having the second strongest
correlation with OrgNO however, it is very low (Pearson's $r = 0.47$).

There is also a substantial drop in LVOOA concentrations in the monsoon period, suggesting a decrease in light
due to cloud cover could be responsible or increased washout from rainfall (Figure 9). In contrast, SVOOA
increases significantly during this period which may be caused by the increase in relative humidity and LWC
leading to increased aqueous-phase aerosol uptake. This is supported by SVOOA having the second strongest
correlation with liquid water content (LWC), after SVBBOA for the combined dataset (all seasons) (Pearson's $r$
$= 0.56$). Furthermore, it has the strongest correlation out of all factors with LWC during the monsoon (Pearson's
$r$ for: COA $= 0.27$, NHOA $= 0.02$, SFOA $= 0.42$, HOA $= 0.04$, SVBBOA $= -0.07$, LVOOA $= 0.24$ and SVOOA
$= 0.49$).

Both SVOOA and LVOOA have the lowest PAH content and their PAH mass spectra are a mixture of peaks also
seen in factors corresponding to primary aerosol composition. LVOOA is more oxygenated and has a large
benzofuran peak (*m/z* 167), along with a naphthoquinone peak (*m/z* 158) which distinguishes it from the other
factors. SVOOA conversely has a mixture of peaks seen in most primary factors but the difference is a particularly
large amount of naphthalene (*m/z* 127 and 128).

The sum of LVOOA and SVOOA is considered the total PM$_1$-SOA and when compared to the total primary OA
(POA = HOA + NHOA + SFOA + SVBBOA + BC), it is lower in concentration during all periods (Figure 9).

### 3.4. Elemental analysis


The measurements of elemental ratios is shown in Figure 14 where the VK diagram for each period can be used
to determine the level of oxidation and chemical functionalisation within the aerosol composition. Here, there is
a similar aerosol composition between the pre-monsoon and monsoon periods which is a consistent theme
throughout these analyses. There are however slight differences in gradient where the data for the pre-monsoon
mainly falls across a gradient of -1. This equates to an increase in carboxylic acid or a simultaneous increase in
carbonyl and alcohol functional groups. The formation of photooxidation products likely explains this because
concentrations of LVOOA are particularly high (Section 3.3.4). The pre-monsoon has periods of low carbon
oxidation states ($\overline{OS_C}$) which can be explained by simultaneous low concentrations in SOA and high primary OA
concentrations. For example, the period ~12/06/18-16/06/18 relates to low concentrations in both SVOOA and
LVOOA and high concentrations in HOA (Figure 8). There is also evidence of higher concentrations of aged



aerosol during the pre-monsoon as higher carbon oxidation states ($\overline{OS}_C$) are observed. This shows higher levels of aging and oxygenation which is likely due to photochemistry (Heald et al., 2010). During the start of the monsoon, the data falls on a gradient of -1 but it gradually increases to a gradient which is similar to the post-monsoon. This shift is most likely due to the increase in SVOOA which suggests it is composed of alcohol and peroxide

functionalities. The same can be said for the post-monsoon where the gradient increase will likely be due to the influence of SVBBOA and SVOOA.

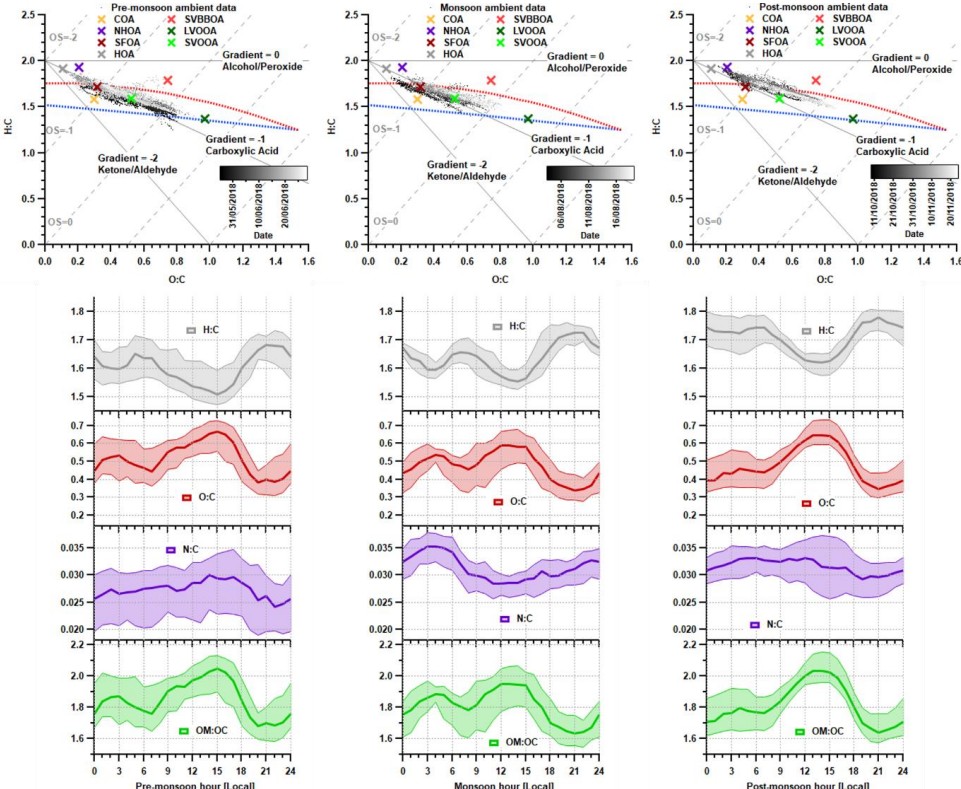

**Figure 14 – Van Krevelen (VK) diagrams and median diurnal cycles for elemental ratios during each measurement**
**period. Each VK diagram contains the H:C vs O:C data for the PMF solutions and the raw AMS measurements which are coloured based on the time of the measurement. The carbon oxidation states ($\overline{OS}_C \approx 2O/C – H/C$) are shown using grey dashed lines and the functional group gradients are shown using solid grey lines. The blue and red dashed lines demarcate the region where published ambient OOA measurements are commonly found (Ng et al., 2011).**

The diurnal profile for H:C almost perfectly mirrors that of O:C and follows the primary factors (HOA, NHOA and SFOA) for all three measurement periods (Figure 10 and Figure 14). The N:C however is highly variable across the three periods with significant changes in its diurnal pattern. During the pre-monsoon, nitrogen is mainly within the SOA fraction as the N:C ratio follows the O:C ratio in the diurnal cycle and peaks in the afternoon with LVOOA (Figure 14). This changes during the monsoon period as there is an early morning (00:00 – 06:00) peak

in N:C. This simultaneous increase in N:C and O:C may suggest that dark oxidation via nitrate radicals is occurring. This is supported by the OrgNO diurnal cycle for the monsoon (Figure S18) which shows a small rise





in the interquartile range during the morning hours (00:00-07:00) and could tentatively be linked to the same rise in N:C. There is also an increase in LVOOA during these hours showing that it is composed of dark oxidation products. A small rise in N:C is also seen during the morning of the post-monsoon however, this is likely flattened

due to the large increases in N-rich factor concentrations, such as NHOA and SVBBOA (Figure 10).

## 4. Discussion

### 4.1. Traffic sources: the existence of nitrile compounds and the separation of NHOA and HOA based on fuel type

The existence of nitrile compounds within NHOA is evident through mass spectral comparisons as explained in Section 3.3.1. VOC measurements also support this observation as the graph in Figure S12 shows correlations between PTR-QiTOF measurements of $C_xH_yN_z$ species and PMF factors. Several of these species have previously been identified as nitrile compounds by Brilli et al. (2014). These species have a stronger correlation with NHOA over all factors which suggests possible partitioning of nitrile VOCs into the particle phase or vice versa. These

compounds have not been identified in published AMS spectra; however, many have published amine peaks within factors. Most studies name these factors nitrogen organic aerosol (NOA) however, some relate to specific local organic aerosol (LOA) (Aiken et al., 2009; Docherty et al., 2011; Sun et al., 2011; Saarikoski et al., 2012; Hayes et al., 2013; Bottenus et al., 2018; Zhang et al., 2018). The main difference between Delhi measured NHOA and literature NOA (or LOA) is the absence of even $m/z$ amine peaks in NHOA such as $m/z$ 56 ($[C_3H_6N]^+$), $m/z$

58 ($[C_3H_8N]^+$), and $m/z$ 84 ($[C_5H_{10}N]^+$). The Van Krevelen (VK) diagram in Figure S13 shows four examples of, what is collectively termed here as NOA, and shows a large variety of elemental compositions. Most studies relate the amine content to secondary sources or specific local sources. Ye et al. (2017) report a NHOA factor which they suggest is also composed of amines however they state this is inconsistent with their findings that NHOA was related to fossil fuel combustion. The NHOA factor they measured is the closest factor within the VK diagram

to NHOA measured in Delhi. They also measured similar odd nitrogen $m/z$ peaks at 41 ($[C_2H_3N]^+$), 43 ($[C_2H_5N]^+$), 55 ($[C_3H_5N]^+$) and 57 ($[C_3H_7N]^+$) and it is therefore suggested that this factor may also contain nitrile compounds. Nitrile compounds within PM are usually considered a sign of nitrogen-rich fuel combustion and these findings provide useful tracers for biomass burning such as acetonitrile (Simoneit et al., 2003; Weimer et al., 2008). However, there appear to be no publications reporting ambient atmospheric nitriles being linked to traffic sources.

High molecular weight nitrile compounds with 7 to 22 carbon atoms ($C_7$-$C_{22}$) have been measured during high emission episodes of ammonia and 'fatty carboxylic' or alkanoic acids (Abas et al., 2004; Ozel et al., 2010; Simoneit et al., 2003). These observations are supported by laboratory studies which show atmospheric ammonia and alkanoic acids react to form alkyl cyanides via an amide intermediate (Simoneit et al., 2003; Zhao et al., 2009). The characteristically high ammonia concentrations in Delhi could therefore make this a viable reaction

pathway (Saraswati et al., 2018; Nakoudi et al., 2019).

Studies that have measured $C_9$-$C_{30}$ alkanoic acids suggest they originate from fossil fuel combustion, biomass burning, cooking and microbial activities (Abas et al., 2004; Kang et al., 2016; Gupta et al., 2018; Gadi et al., 2019). A substantial amount has been shown to be emitted from diesel engines where the majority originates from the engine lubricant (Lim et al., 2015). Emissions also increase with the use of biodiesel which is becoming a

common fuel-type in Delhi (Cheung et al., 2010). Two previous studies measured $PM_{10}$ composition at IGDTUW





and found a large fraction of the organic aerosol is composed of n-alkanoic acids (Gupta et al., 2018; Gadi et al., 2019). They suggest that cooking and traffic emissions influence the n-alkanoic acid abundance and their concentrations are higher during the post-monsoon and winter months. This is a similar trend to NHOA and there are numerous studies which show traffic to be a large source of ammonia emissions in Delhi, different cities in

India, and in different cities worldwide (Sharma et al., 2014; Sun et al., 2017; Elser et al., 2018; Saraswati et al., 2018). It is therefore possible that the nitrile compounds within NHOA are formed through the reaction of alkanoic acids and ammonia.

The large differences in the PAH composition of NHOA and HOA described in Section 3.3.1 is a key characteristic which separates these two traffic factors. From the *m/z* 100-320 mass spectra in Figure 7 it is clear

that masses are much higher in the later part of the NHOA spectrum than for HOA. These include (but are not restricted to) higher molecular weight UnSubPAHs and MPAHs (Figure 12) and these are diagnostic markers for a diesel source (Singh et al., 2010). This suggests that HOA may originate from less dense traffic fuel types that have much lower aromatic content such as gasoline and compressed natural gas (CNG). What weakens this hypothesis is the absence of NHOA in the pre-monsoon and monsoon periods as diesel vehicles and trains are in

operation throughout the year, although the additional contribution from diesel generators may vary throughout the year. This could, therefore, suggest NHOA is influenced by meteorology and forms during favourable conditions for aerosol partitioning in the post-monsoon. These conditions allow for the partitioning of more volatile fractions of NHOA, such as the nitriles. Diesel vehicles are not normally permitted on the roads directly adjacent to the measurement site. Thus, the low boundary layer post-monsoon may also result in more efficient

advection and build-up of diesel emissions to the measurement site at low height, whilst emissions may be more efficiently dispersed during the other seasons.

Similar UnSubPAH and MPAH species in NHOA have been observed in laboratory studies measuring PAHs from diesel engines, but the ratios of these species largely depends on the mix of biodiesel and the running conditions of the engine (He et al., 2010; Tsai et al., 2011; Zheng et al., 2017). There is, however, a study which

took place in Taj, India, which measured both indoor and outdoor PAH concentrations in homes along roadsides or in residential areas. Through principal component analysis, the authors were able to separate factors based on different fuel types (Masih et al., 2012). At all measurement sites, factors were resolved relating to diesel and others relating to petrol and CNG. The latter is widely used in Delhi and is mandatory for public transport vehicles (taxis, moto-rickshaws). The diesel-related factors were high in acenaphthylene (*m/z* 151 and 152), acenaphthene

(*m/z* 153 and 154), fluorene (*m/z* 165), anthracene and phenantharene (*m/z* 178) and, carbazol (*m/z* 216 and 217), which is consistent with NHOA. The petrol and CNG-related factors were also consistent with HOA where methyl-naphthalene (*m/z* 141 and 142) and dimethyl-naphthalene (*m/z* 155 and 156) were particularly high and these are the distinguishing peaks in the PAH spectrum for HOA. The evidence gathered therefore suggests that NHOA and HOA are separated based on fuel-type, where NHOA is influenced by diesel emissions and HOA by

gasoline and compressed natural gas. The separation of factors based on fuel-type is not often observed due to the almost identical diurnal patterns they often share. In Delhi however, this separation is most likely possible due to the restrictions of heavy goods vehicles during the day (07:00-23:00) which increases diesel emissions at night.

There is a strong correlation between NHOA and the dibenzodioxin fragment ion, benzodioxan ($[C_8H_8O_2]^+$) (Pearson's $r = 0.80$), and this is consistent with dibenzodioxin peaks seen in its PAH spectrum (Figure 12). There

is also a mild correlation with chloride (Pearson's $r = 0.62$) which could be explained by the presence of





polychlorinated dibenzodioxins (PCDD). It is well known that diesel combustion is a source of PCDDs and therefore could be a form of organic chloride associated with NHOA (Laroo et al., 2012; Wang et al., 2012). However, NHOA could also be similar in volatility to ammonium chloride which could also explain this.

Traffic-related organic aerosol (NHOA+HOA) amounts to the largest primary source of $PM_1$ across the pre-
monsoon and monsoon periods (Figure 9). It is also the joint second highest source (with SOA) during the post-monsoon period which suggest that a reduction in its emissions will result in a drop in $PM_1$ organic aerosol across the majority of the year. This conclusion is consistent with the analysis of unit mass resolution measurements from this and other AMS instruments operated in Delhi (Reyes-Villegas et al., 2020), which also found traffic to be the largest contributor to $PM_1$. Targeting emissions from traffic can therefore be viewed as the highest priority
when developing air quality mitigation strategies for $PM_1$, especially considering that traffic sources also supply some of the precursor gases for the formation of the SOA components.

### 4.2. Sources of cooking activities and their link to OrgNO

The COA time series has a moderate correlation with the AMS measured $[C_6H_{10}O]^+$ peak (Pearson's $r = 0.65$)
which is often used as an internal tracer for cooking related factors (Sun et al., 2016). This tracer has previously been used on AMS measurements taken in Western and East Asian countries. The reason for the reduced correlation in this study may be a difference in the style of cooking in Delhi and/or the contribution of other sources to this peak. Western and East Asian cooking is shown to produce short-chained organic acids through rapid and shallow frying (Reyes-Villegas et al., 2018a). In Delhi, however, food is generally cooked on a low heat
and over a longer period of time. It is therefore hypothesised that longer-chained oxygenated fatty acids are associated with cooking activities in Delhi. This is supported by measurements taken at IGDTUW showing long-chained fatty acids being linked to cooking sources (Gupta et al., 2018; Gadi et al., 2019).

This difference in cooking style might also explain why the mass spectrum has an uncharacteristically large peak at $m/z$ 44 $[CO_2]^+$ which is considered to be mainly due to carboxylic acid groups (Duplissy et al., 2011; Ng et al.,
2011). Long-chains of $C_xH_y$ atoms along with an increased number of –COOH groups fragment in the AMS resulting in mainly aliphatic ions and increased $m/z$ 44. As a result, the COA factor has a particularly large O:C ratio (0.33) when compared to literature values (Table S2) which pulls it towards a carboxylic acid composition gradient on the VK diagram in Figure S13. Additionally, some of the strongest correlations of VOCs with COA in the lower $m/z$ part of the PTR-QiTOF spectrum relate to the $C_xH_yN_z$ and $C_xH_yO_z$ families (Figure 13). These
observations also support COA being composed of long-chained carboxylic acids and could also suggest that COA is influenced by the oxidation of organic acid VOCs.

The $C_xH_yN_zO_t$ family in both the AMS and PTR-QiTOF mass spectra are most closely correlated with COA (Figure 13). This coincides with the OrgNO having the strongest correlation with COA, out of all PMF factors. Furthermore, the COA spectrum for the inorg-org combined PMF solution shows a large $NO^+$ ($m/z$ 30) peak with
no associated ammonium peaks which implies the nitrate is organic (Figure S15). The evidence gathered, therefore suggests emissions associated with food preparation in Delhi is nitrogen rich and contributes the most to OrgNO concentrations. The presence of N-containing aromatic compounds (NACs) in $PM_{2.5}$ from wood and charcoal fuelled cooking stoves has also recently been measured experimentally (Xie et al., 2020). Again, the majority of the NACs identified in these measurements were of the $C_xH_yN_zO_t$ family.





COA could also be influenced by the emissions of fuels used for food preparation as this would correlate highly in space and time. Äijälä et al. (2017) published a similar COA spectrum containing high $m/z$ 44 (and $m/z$ 28) peaks and found their measured COA is influenced by biomass burning. This could also be true for Delhi as residents, roadside stalls and restaurants (e.g. Dhabas) cook on open fire pits using fuels such as wood and dung cakes, and would explain a mixed factor profile of COA with BBOA influence.

The strong association of PAHs with cooking activities is well established and is an important component of cooking aerosol to consider due to its carcinogenicity (Svedahl, 2018; Lin et al., 2019). As explained in Section 3.2.2, COA has strong correlations with multiple AMS measured PAH fragments (Figure 13) and a particularly high peak at $m/z$ 91 $[C_7H_7]^+$ within its mass spectrum (Figure 7). This shows that UnSubPAHs are a defining part of a COA signature. As the COA spectrum has strong similarities with SVOOA spectra (due to $m/z$ 44), it's high

PAH composition could help future studies. For AMS measurements in Delhi, $m/z$ 91 could be a useful indicator of COA along with the $m/z$ 55:57 ratio for studies that do not quantify $m/z$ >120.

Unlike other source factors, COA does not significantly increase in concentration during the post-monsoon (Figure 9). Cooking activities are therefore relatively constant across the year in Old Delhi although there is a substantial drop in COA during the monsoon. This is likely because of the increased rainfall and RH (monsoon

average ~76 %) causing increased rates of below cloud scavenging or washout. The propensity of this happening to COA over other organic aerosol could be down to composition as its high fatty acid content will increase its hygroscopicity, making it more likely to be scavenged. This could explain the reason for the low correlation with LWC (Pearson's $r = 0.15$) as an increase in water content and RH will most likely increase wet deposition. An alternate reason could simply be due to a decrease in outside cooking activity as more residents likely cook indoors

to escape the rain.

### 4.3. Sources of burning

The results indicate the two burning-related PMF factors SFOA and SVBBOA are fresh and aged organic aerosol, respectively. The higher afternoon concentrations of SVBBOA and a high signal at $m/z$ 44 are key indicators of

its lower volatility and oxidation. This is also consistent with the analysis of AMS flux measurements taken during the post-monsoon (Ben Langford pers. commun.). From the analysis, two burning factors were resolved where one had higher emissions during the morning and evening suggesting it to be localised and a primary source. The other factor showed a lower and broader emission in the morning suggesting it is secondary. A similarly aged PMF factor to SVBBOA, named oxidised-BBOA, was also resolved from AMS measurements in Kanpur. They

suggest this O-BBOA mainly formed through photochemical oxidation of primary emitted BBOA in the local area (Chakraborty et al., 2018). This may be the case in Delhi; however, there are peaks in the time series of SFOA which coincide with an absence of SVBBOA, particularly during peak photochemical conditions in the pre-monsoon (Figure 8). The origin of this source therefore may not be local to Delhi and instead the aerosol could have travelled from further afield.

The contribution of SVBBOA to the total $PM_1$ mass is insignificant during the pre-monsoon (~1%) and monsoon (~0%) periods. However, during the post-monsoon it is the second largest contributing factor (~12%) (Figure 9). This suggests a specific source that only occurs during the post-monsoon period. The most likely cause is large-scale crop residue burning after paddy fields are harvested in October and November which has been widely





blamed for poor air quality in Delhi (Liu et al., 2018; Jethva et al., 2019; Beig et al., 2020; Mukherjee et al., 2020).

In Figure 15, the SVBBOA time series is compared with fire counts data from the NASA Visible Infrared Imaging Radiometer Suite (VIIRS) sensor on the Suomi National Polar-orbiting Partnership (S-NPP) satellite (available at: https://earthdata.nasa.gov/earth-observation-data/near-real-time/firms). The daily average fire counts are taken from a custom polygon of the Haryana region from which the city of Delhi itself has been removed. The strong correlation of SVBBOA with fire counts (Pearson's $r = 0.73$) suggests the source of this factor is mainly regional

crop residue burning outside of Delhi. Whilst other factors, and particularly SFOA, also correlate significantly with the fire counts (as a result of all concentrations increasing post-monsoon), the PMF analysis identifies some SFOA but very little SVBBOA during monsoon (Figure 8 and Figure 9). This suggests that the distinction between SFOA and SVBBOA is fairly robust in quantifying the contribution of crop residue burning to Delhi's $PM_1$.

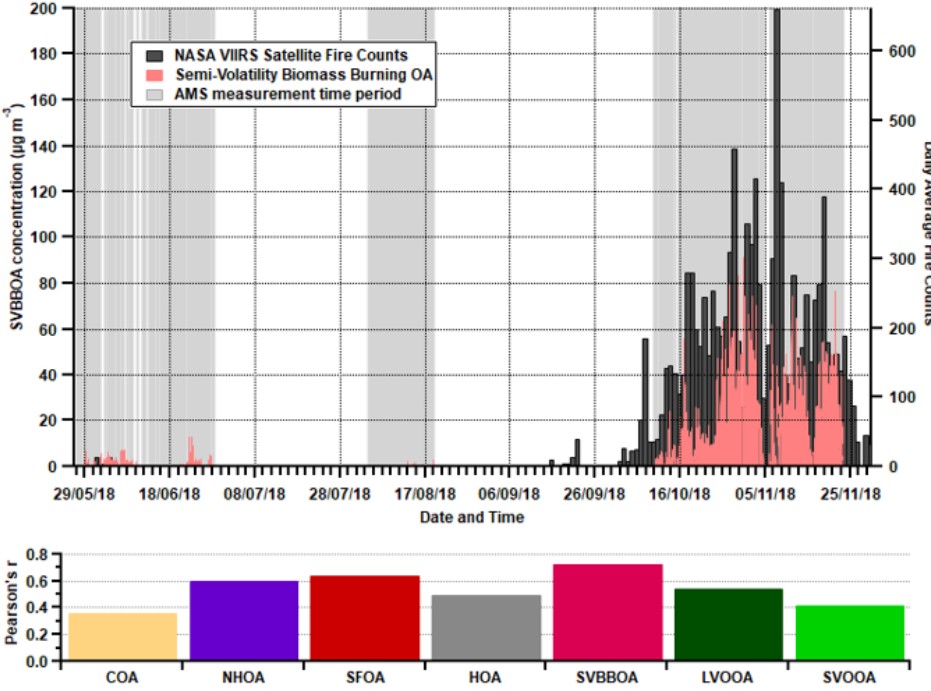

**Figure 15 – Upper panel: time series for SVBBOA along with the daily average fire counts from the NASA Visible Infrared Imaging Radiometer Suite (VIIRS) sensor on the Suomi National Polar-orbiting Partnership (S-NPP) satellite. Lower panel: Pearson's r correlations between daily average factor concentrations and the daily average fire counts. The VIIRS data covers all days displayed along the x-axis while the AMS measurements cover the grey shaded regions.**

These findings show that crop burning is a major source of air pollution during the post-monsoon period when PM concentrations are at their highest. In 2018 a government initiative was introduced to improve air quality in Delhi by encouraging farmers outside the national capital region (NCR) to supply their agricultural waste for conversion to bioenergy (Bhuvaneshwari et al., 2019). Our measurements reveal that despite this intervention, the burning of crop residues in the post-monsoon still accounted for ~20% of the total $PM_1$ increment over the pre-





monsoon period. Our measurements suggest that in 2018 there was still scope for further reductions from the
        agricultural sector.

        Studies have observed $KNO_3$ formation in aged smoke from biomass burning via the reaction of KCl with gaseous
        $HNO_3$ (Li et al., 2003; Wang et al., 2017). A subsequent substitution reaction with $NH_3$ forms $NH_4NO_3$ and this
        formation pathway could explain why SVBBOA has the strongest correlation with nitrate and ammonium (Figure

9). SFOA, however, is weakly correlated with nitrate which suggests it may not be linked to crop residue or wood
        burning. The defined morning and evening peaks in the SFOA diurnal cycle, and a less-oxidised composition
        shows SFOA is more likely to be linked to freshly emitted local area sources. It could, therefore, be due to
        municipal waste burning, which is a common practice in Delhi, particularly during the morning and evening
        (Nagpure et al., 2015). Municipal waste burning is also often found to produce higher PCDD and PCDF

concentrations than biomass burning or traffic sources (Lavric et al., 2004; Chakraborty et al., 2013; Verma et al.,
        2016; Zhang et al., 2017; Stewart et al., 2020). The high dibenzodioxin and dibenzofuran peaks in the SFOA mass
        spectrum (Figure 12) coincide with strong correlations with our proposed novel-tracers for PCDDs and PCDFs,
        benzodioxan $[C_8H_8O_2]^+$ and furan $[C_4H_4O]^+$ (Figure 9). This shows SFOA has the strongest link to PCDDs and
        PCDFs which suggests a substantial fraction of its mass may be due to the burning of highly chlorinated municipal

waste.

        Although the results suggest strong links between SFOA and municipal waste burning, it is likely that this factor
        contains other solid fuel burning sources as well. The high correlation with acetonitrile (Figure 9) and mass
        spectral peaks at $m/z$ 60 and 73 (Figure 7) show that SFOA includes a significant amount of organic aerosol from
        biomass burning. This will most likely include emissions from solid fuels such as dung cakes and wood which are

used for cooking and warmth in Delhi (Sen et al., 2014; Pervez et al., 2019).

        The measurement site is also positioned near two cremation sites (~500-750 m) in the east and southeast along
        the Yamuna river where open funeral pyres and furnaces are used frequently. The largest of the two is directly
        east and coincides with frequent peaks in SFOA and PAH concentrations (Figure 11). Studies have shown that
        funeral pyres in India emit organic carbon and organic aerosol on a comparable level to transport emissions, where

it was estimated to be equivalent to ~10-23% of organic aerosol mass emitted from regional fossil fuel and biofuel
        burning (Chakraborty et al., 2013; Pervez et al., 2015). It is also shown that funeral pyres emit a significant amount
        of PAHs that are particularly high in benzene ring count (Dewangan et al., 2014) which is consistent with the high
        peaks later in the PAH mass spectrum for SFOA (Figure 12). SFOA is therefore also likely to include aerosol
        emitted from funeral pyres.

The large south-easterly peaks in the polar plots for SFOA, COA, NHOA and PAH, are concurrent which suggests
        a common source (Figure S10). These peaks occurred during the late hours of the night (~10-12 p.m.) and on
        three consecutive occasions which indicates the source is infrequent. They also occur during the Diwali festival
        and it is likely that they are due to pyrotechnical activity. A possible explanation is the PMF algorithm is unable
        to resolve these peaks into a single factor due to their infrequency. Further AMS studies on pyrotechnical activity

may help to provide characteristic mass spectra which could be used, along with ME-2, to separate factors on
        occasions such as this.

        Combining the two factors, SFOA and SVBBOA, shows that organic aerosol associated with burning-related
        sources contributes the highest $PM_1$ mass during the post-monsoon period (Figure 9). This is coincidentally the
        period when concentrations are the largest by a significant amount, in part, due to accentuated boundary layer



dynamics. The post-monsoon is therefore a critical period which detrimentally decreases the health of Delhi's population. Combustion sources, including crop residue, municipal waste, funeral pyre cremations and solid fuel burning, therefore need to be considered when creating new air quality mitigation strategies.

### 4.4.   Sources of chloride

The concentrations of chloride in Delhi are particularly large relative to levels reported in other megacities, and a number of studies have linked it to biomass burning and industrial activities (Stone et al., 2010; Gani et al., 2019). There are possible industrial sources of chloride to the northwest of Delhi and these include steel pickling and electronic waste recycling sites that are suggested to emit gaseous HCl (Gani et al., 2019). The same early morning chloride peak reported by Gani et al. (2019) was measured at IGDTUW and occurs around 07:00-08:00 (Figure

3). This morning peak is however also seen using a HR-TOF-AMS in Kanpur, India, which suggests that this may not be linked to the same industrial sources (Chakraborty et al., 2018). Furthermore, there is no particular wind directional preference for the morning peak at IGDTUW (Figure S9) whereas Gani et al. (2019) show a preferred North Westerly wind. Our findings could instead be consistent with a nocturnal area source of HCl which preferentially partitions into the particle phase at night, accumulates in the shallow nocturnal boundary layer, and

eventually evaporates in the morning. There is also no evidence to suggest the steel pickling and electronic waste recycling industries are more active during the post-monsoon and winter months when chloride concentrations are at their highest.

In both Delhi and Kanpur residents burn refuse either for warmth or to clear the streets and this is likely to be a significant source of chloride. Three Indian-based studies show that aerosol emitted from burning waste is

composed of chlorine associated PAHs formed mainly from plastics such as polyvinylchloride (PVCs) and polychlorinated biphenyls (PCBs) (Chakraborty et al., 2013; Vreeland et al., 2016; Shivani et al., 2019). At IGDTUW, chloride has a strong correlation with the PAH time series (Pearson's $r = 0.72$) which is significantly stronger than that of nitrate (Pearson's $r = 0.58$), sulphate (Pearson's $r = 0.21$) and ammonium (Pearson's $r = 0.59$). The diurnal cycle for PAHs also shares a similar temporal behaviour in the morning to chloride where the

median peaks at ~07:00-08:00. This coincides with the regular occurrence of residents sweeping the previous days waste into piles and burning them. One study indicates that this behaviour results in larger amounts of municipal waste being burnt in the morning compared to the evening (Nagpure et al., 2015).

SFOA is strongly related to both PAHs and chloride and although SFOA concentrations in the pre-monsoon and monsoon are comparatively low (Figure 9), they are still comparable with chloride concentrations. The link with

burning plastics through PCDDs and PCDFs further supports this and the results gathered therefore suggest that municipal waste burning may be the largest source of $PM_1$ chloride during the pre-monsoon and monsoon periods. Another possible source of chloride is from the annual crop burning that occurs during the post-monsoon period. It is well known that wood burning is a strong source of particulate KCl (Li et al., 2003; Lanz et al., 2008; Weimer et al., 2008) and while KCl itself is not quantifiable by the AMS, it goes on to form $NH_4Cl$ via the intermediate

reaction of HCl with $NH_3$ (Sullivan et al., 2007; Wang et al., 2017). SVBBOA has a strong correlation with chloride (Figure 9) and the additional results from the second (Org-Inorg) PMF analysis, which included the inorganic components, associates the majority of the post-monsoon ammonium chloride with SVBBOA (Figure S15). There are many studies which support this conclusion and show crop residue burning to be a significant





source of chloride (Li et al., 2003; Sullivan et al., 2007; Lanz et al., 2008; Weimer et al., 2008; Christian et al., 2010; Wang et al., 2017). However, how much of this association is down to the PMF algorithm resolving similarities in thermodynamics and thus reflect the volatile nature of both components is uncertain. Furthermore, SFOA has a strong correlation with chloride which suggests localised burning is an equally significant source. Crop burning therefore likely explains a significant fraction of the high levels of chloride in the post-monsoon while municipal waste burning is likely the dominant source during the pre-monsoon and monsoon periods.


### 4.5. Secondary organic aerosol

The two SOA factors LVOOA and SVOOA compare well with literature in the VK diagram (Figure S13) and they lie in areas which are consistent with their respective aged and less-aged SOA profiles. When compared to a few AMS studies measuring PM from around the world, only measurements in China gave an LVOOA factor
with a larger O:C ratio (Table S3). This reflects how oxidised LVOOA is in Delhi and suggests a large amount of atmospheric processing. LVOOA is influenced mainly by photochemical oxidation as suggested by its afternoon peak (Figure 10) and its correlations with oxidised VOCs (Figure 13). SVOOA however may instead be more influenced by gas-aerosol partitioning and aqueous-phase aerosol uptake as it has a higher correlation with LWC (Pearson's $r$ = 0.56, Figure 9).

The SVOOA mass spectrum is similar to the primary factor, COA, shown by a similarly high $m/z$ 55:57 ratio (2.87) (Figure 7). They also share a similar affinity with OrgNO as both have the strongest correlations with OrgNO (Figure 9) and similar correlations with $C_xH_yO_zN_t$ VOC species (Figure 13). This is supported by the Inorg-Org PMF solution giving similar COA and SVOOA spectra, which includes both having a high $NO^+$ (30 $m/z$) peak (Figure S15). The largest difference between the two factors is the level of oxygenation where SVOOA
has an O:C ratio of 0.52 and could suggest SVOOA may be formed via oxidised COA. The pollution rose plots support this as both COA and SVOOA show a similar wind directional preference (Figure 11). This may also explain the simultaneous drop in COA and increase in SVOOA during the monsoon period as COA is oxidised through aqueous-phase chemistry to form SVOOA.

The diurnal cycles for N:C and O:C (Figure 10) appear to show a close relationship during the pre-monsoon
indicating that nitrogen content is being driven by photochemistry. This suggests that the organic nitrogen species are mainly of the $C_xH_yN_zO_t$ family rather than the $C_xH_yN_z$ family during this period. A similar pattern is also seen in Kanpur where the N:C ratio closely follows the O:C ratio (Chakraborty et al., 2016a). There is however an increase in N-rich factors such as NHOA, SFOA and SVBBOA during the post-monsoon period which breaks the similarities in N:C and O:C diurnal cycles (Figure 10). NHOA and SVBBOA are shown to contain nitrogen in the
form of nitrile and/or amine compounds which are of the $C_xH_yN_z$ family (see Sections 4.1 and 4.3). This shows the post-monsoon nitrogen species composition is a mixture of $C_xH_yN_zO_t$ and $C_xH_yN_z$ chemical families. During the monsoon however, the concentrations of N-rich factors (NHOA, SFOA and SVBBOA) are low which removes their influence on N:C during the night. This allows for the N:C ratio to be driven by dark oxidation via nitrate radicals which is seen in the early morning peak (Figure 14). The overall contribution of dark oxidation to SOA
formation however is relatively small in Old Delhi. Evidence of which is shown by the coinciding rise in LVOOA concentrations which is small when compared to the following photochemically produced afternoon peak (Figure 10).



### 5. Conclusions

PM$_1$ measurements were taken using a HR-TOF-AMS in Old Delhi for the first time, covering three seasons (pre-monsoon, monsoon and post-monsoon). Results show large concentrations of aerosol species, particularly during the post-monsoon when ammonium and organic aerosol increase by a factor of ~2-3, and chloride and nitrate increase by a factor of ~5-6. These high post-monsoon concentrations have been linked to an increase in burning emissions mainly from crop residue and solid fuel.

A 7-factor solution was resolved from source apportionment analysis using PMF and this included two traffic-related factors (NHOA and HOA), two burning-related factors (SFOA and SVBBOA), a cooking factor (COA) and two SOA factors (LVOOA and SVOOA). NHOA is separated from HOA by the presence of what we believe to be nitrile species within its composition and their existence is previously unpublished in AMS literature. These nitrile compounds are hypothesised to form from the reaction of alkanoic acids and ammonia. Using a unique

view of AMS factor PAH peaks, we establish that NHOA, unlike HOA, also includes higher mass PAHs. This suggests NHOA is heavily influenced by diesel engine emissions and coincides with the easing of heavy-goods vehicle restrictions during the night. HOA, however, was characterised by low molecular weight PAHs including key PAH species markers which suggest it, instead, originates from CNG and petrol engine emissions.

The existence of OrgNO species is evident with concentrations peaking at ~ 25 µg m$^{-3}$ during the Diwali festival

when large-scale pyrotechnical activity occurred. It was also found that COA is closely related to OrgNO through correlations with OrgNO and C$_x$H$_y$N$_z$O$_t$ PTR-QiTOF ions. This may suggest that in Delhi OrgNO is formed mostly from cooking activities.

The same associations with OrgNO are also seen with the secondary SVOOA factor, which shares similarities in its mass spectrum to COA. This suggests it is formed from oxidised COA. The simultaneous loss of COA and

increase in SVOOA during the monsoon period also indicate aqueous-phase oxidation occurs. LVOOA, however, is shown to form mainly through photochemistry but a small contribution is also made through dark oxidation via nitrate radicals.

The COA mass spectrum was found to have an unusually high degree of oxygenation. This may be explained by the difference in cooking style in Delhi as food is cooked slowly and on a low heat which emits long-chained

oxygenated fatty acids. PAHs are also shown to be an important component of COA composition, and it contributes a large proportion to the total PAH mass. A particularly high PAH fragment ion [C$_7$H$_7$]$^+$ peak (*m/z* 91), along with the literature established high *m/z* 55:57 marker, could be used as an additional indicator of COA in Delhi for future AMS studies, especially those measuring below *m/z* 120.

Chloride showed the largest relative increase out of the four inorganic species and organic aerosol, increasing

480% from an average of 1.3 µg m$^{-3}$ in the pre-monsoon to 7.6 µg m$^{-3}$ post-monsoon. Two burning-related factors, SFOA and SVBBOA, both show strong associations with chloride for different reasons. Using earth observations, SVBBOA is tightly linked to regional crop residue burning and is more secondary in character. A similar spectrum to SVBBOA was resolved using an alternative inorganic-organic PMF analysis but with large ammonium chloride peaks. This suggests that considerable amounts of ammonium chloride are linked with crop residue burning. We

introduce novel AMS measured tracers which show the presence of PCDBs and PCDFs. This offers a new way to associate sources to plastic or Cl-rich fuel burning. Using these tracers, SFOA was found to have significant



links to municipal waste burning. With SVBBOA concentrations being insignificant during the pre-monsoon and monsoon periods, municipal waste burning is therefore likely the most significant source of chloride during the pre-monsoon and monsoon periods. During the post-monsoon, however, it is concluded that both crop residue

burning, and municipal waste burning are key contributors to the large increase in chloride.

SFOA was also linked to other sources of solid fuel burning such as wood and dung cakes which are commonly used in Delhi, particularly during the post-monsoon (and winter) when Delhi residents burn solid fuel to keep warm. Wind directional data also suggests funeral pyres situated east of the measurement site likely contribute to SFOA as frequent excursions point in this direction.

Overall, the burning related sources (SFOA+SVBBOA) are the largest primary source of $PM_1$ during the post-monsoon. The high concentrations of SVBBOA suggest that there is further scope to mitigate $PM_1$ concentrations through strategies aimed at reducing crop residue burning. This, combined with reductions in other solid fuel burning sources, could significantly decrease the large peaks in $PM_1$ concentrations during the post-monsoon. However, overall, our measurements suggest that reducing traffic emissions will have the greatest reduction on

$PM_1$ organic aerosol concentrations across the majority of the year, as total traffic aerosol (NHOA+HOA) is shown to be the highest primary contributor during the pre-monsoon and monsoon.

### Acknowledgements

This work was supported by UK NERC project DelhiFlux under the Newton-Bhabha Fund Programme "Air

Pollution and Human Health in a Developing Megacity (APHH-India)", NERC reference numbers: NE/P016502/1 and NE/P016472/1. The NERC National Capability award SUNRISE (NE/R000131/1) supported the monsoon measurements and James Cash is supported by a NERC E[3] DTP studentship (NE/L002558/1). TKM is thankful to Director CSIR-National Physical Laboratory for allowing us to carry out this research. Authors gratefully acknowledge the financial support provided by the Earth System Science Organisation, Ministry of

Earth Sciences, Government of India under the Indo-UK Joint collaboration vide grant no MoES/16/19/2017-APHH (DelhiFlux) to conduct the research. The paper does not discuss policy issues and the conclusions drawn in the paper are based on interpretation of results by the authors and in no way reflect the viewpoint of the funding agency.

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
