# Peer review of "Seasonal analysis of submicron aerosol in Old Delhi using high resolution aerosol mass spectrometry: Chemical characterisation, source apportionment and new marker identification"

_Atmospheric Chemistry and Physics, 2020_

## Referee Comment (RC1) · Anonymous Referee #1 · 1 Feb 2021

The manuscript titled "Seasonal analysis of submicron aerosol in Old Delhi using high resolution aerosol mass spectrometry: Chemical characterisation, source apportionment and new marker identification" describes the measurements carried out in the year 2018, in Old Delhi, India, covering 3 seasons: pre-monsoon ($\sim$1 month), monsoon ($\sim$15 days) and post-monsoon ($\sim$6 weeks). The study uses positive matrix factorization (PMF) to interpret the measurements and finds a 7 factors solution that separates cooking organic aerosols (COA), solid fuel OA (SFOA), hydrocarbon-like organic aerosols (HOA), nitrogen-rich HOA (NHOA), semi-volatile biomass burning OA (SVB-

[Figure]

BOA), semi-volatile OA (SVOOA), and low-volatile OA (LVOOA). The Authors find that the major contributor for the PM1 in the area is: 1) sulfate, LVOOA, and COA during the pre-monsoon period; 2) sulfate, SVOOA, LVOOA and HOA during the monsoon period; and 3)LVOOA, SVBBOA, SFOA, and HOA during the post-monsoon period. Traffic and funeral pyres, crop residue, and waste burning are fund to be major contributors. Another important finding is that the concentrations during the post-monsoon period are around 3-fold higher than in the pre-monsoon and monsoon period and that the chloride fraction has a 5-fold increase in the same period. The authors present a large amount of data both in the manuscript and in the supplemental information (SI). They present them clearly and use the SI effectively to show the reader the process that has led them to choose a particular PMF solution using multivariate fits to external tracers. The article presents data and findings that are of interest to the scientific community and in a location where air pollution is very high and affects a large number of people. The methods used are sound. The interpretation of the data and the conclusions are well rooted in the data with minimal speculation. The presentation of the data and the results are good, although it could use some more clarity especially in the figures as mentioned below in more detail. The location where the measurements were carried out presents a number of challenges such as high temperatures and high relative humidity that can be really tricky for instruments. I think the authors have generated a great dataset in those challenging conditions. The authors also did an excellent job at interpreting such a complex mixture of sources.

The manuscript is of high quality and within the scope of the journal. I recommend the publication after minor revisions.

Detailed comments:

1- The manuscript similarly to many manuscripts based on AMS data and PMF, makes extensive use of acronyms. These acronyms are probably very familiar to the authors and to experienced AMS users, however, they tend to be hard to follow for readers less involved with AMS and PMF analysis. I recommend making a list of acronyms to help

the reader follow the text.

2- Figure 1. I suggest adding the time series of the standard AMS species (NO3, SO4, NH4, Chl, and Org) and BC. This will give the reader a good bird's eye view of the dataset, maybe merging it with some version of Figure 5.

3- Figure 3. I suggest adding dark and light hours with a shaded area for the transition/changing light conditions over the measurement period to help the reader form a picture of the data presented.

4- A lot of figures have tiny labels that are really hard to read especially once the manuscript is printed. In Figure 4 the percentage numbers in each panel are very small I suggest using fewer vales and a larger font. Also, "mean" and "calm" (panels a-e), as well as the Wind Speed values in panel (f) are almost not readable in the printed version.

5- Figure 5. I suggest merging it with figure 1 as mentioned in comment 2

6- Figure 6. Dates are very small, please increase the font.

7- Figure 7 the y axis could be harmonized. It's ok to keep a different scale but I suggest keeping the same number of ticks.

8- Figure 10. the y axis labels for the O:C, H:C and N:C ratios panel are too small. Please reduce the number of ticks, decide how many to put there (3?) and maybe increase the font

9- Figure 11. "mean" and "calm" almost not readable

10- Figure 12. Panel (a) the numbers of ticks could be harmonized by making it the same (4?).

11- Figure 14. I recommend increasing the resolution for the top panel (VK diagrams) the dots are lost even in the electronic version if zoomed in.

12- In the abstract, I recommend adding a mention that sulfate is the largest mass fraction for the pre-monsoon and monsoon periods.

In the methods section

13- at lines 129-131 the Authors mention that they calibrated the AMS "throughout the campaign". I recommend adding a sentence explaining how many times and when (e.g., before, middle, and after?).

14- At lines130 to 135 the Authors mention that in their analysis they had to use different CEs to match the "PM2.5 filter measurements". I recommend expanding this sentence explaining which measurements they are referring to, carried out by which group, with which instrument, and at what time resolution.

15- At lines 146 -148 the sentence "Therefore, only peaks which significantly improved the open and closed signal residuals were fitted regardless of the residuals in the difference (diff = open – closed) signal." is unclear and leaves the reader wonder which peaks were not included. I understand that the fitting at higher m/z is tricky, but I am wondering if the authors can modify or expand on the sentence to clarify the process to the reader, maybe explaining which peaks were not fitted and why.

16- Lines 163-165: ". . . black carbon (BC) measurements which were taken using an Aethalometer AE-31 and corrected for by a Single Particle Soot Photometer (SP-2; Droplet Measurement Technology, Boulder, CO) (Reyes-Villegas et al., 2020).". This sentence is quite vague. I understand that there is a reference to look up, however, I recommend adding a short sentence giving a few more details, e.g., explaining briefly 1)how the Aethalometer data were corrected 2) if/when and for how long the SP2 was co-located with the Aethalometer. Results

17- Lines 235-238: here and in a few other parts, the authors cite "personal communication with Ben Langford". In all those cases I think that this information should be removed as it doesn't seem critical to the point of the sentences unless a paper has

been published in the meantime and can be properly referenced.

18- Line 458: "UnSubPAHs" acronym not defined Conclusions

19- Lines 894-895: "These high post-monsoon concentrations have been linked to an increase in burning emissions mainly from crop residue and solid fuel." Are higher concentrations only due to an increase in burning emissions or the boundary layer height affect these concentrations as well? If that's the case, I recommend adding it here. Supplementary Information

20- Figure S2 y-axis label too small

21- Page 12: "Error! Not a valid bookmark self-reference." Should be "Table S2"

22- Figure S8: "Polar graphs showing the concentrations . . ." add units of concentrations.

23- Figure S9 "Mean" and "calm" not legible.

24- Figure S13: add that the points not labeled neither "Delhi" nor "Chack2018" come from Table S3.

25- Page 20: "The factor mass profiles and their diurnal cycles during each measurement period are summarized in Figure S14". I think it should be "Figure S15"

26- Figure S15 and S16: y-axis labels are too small

Finally, reading the manuscript I have been wondering why the Authors decided not to run the PMF in bootstrap mode for the 7 solutions combined periods.

---

## Referee Comment (RC2) · Anonymous Referee #2 · 16 Feb 2021

General Comments:

The manuscript by Cash et al. describes a chemical characterization and source apportionment work using high resolution aerosol mass spectrometry (HR-AMS) data collected in one of the most polluted and populated areas in Old Delhi, India in 2018. It covered 3 time periods: pre-monsoon (around 1 month), monsoon (around 15 days), and post-monsoon ( around 6 weeks). The study identified two traffic-related factors (NHOA and HOA), two burning-related factors (SFOA and SVBBOA), a cooking factor (COA), and two SOA factors (LVOOA and SVOOA) using positive matrix factorization

(PMF). In the end, the Authors suggest developing air quality policies by mitigating emissions from crop residue burning, open waste burning, and traffic activities. Cash et al. were able to reveal the mystery of high chloride concentration using HR-AMS, which has strong links with the burning sources, especially in the post-monsoon period. In addition, with the rather well-resolved PMF solution, it is certainly a valuable reference for AMS users in the future (e.g., the mass spectra of NHOA, SFOA, SVB-BOA). Collecting and analyzing such a dataset from a high temperature, high relative humidity environment remains challenging, but the authors were able to justify their results using both independent and internal tracers. The multilinear regression analysis is a useful tool for this dataset considering there is no reference HR mass spectrum available for a unique environment, like old Delhi. I believe the contribution of this work is significant and meets the scope of the ACP. Overall, the English are perfect, but the whole manuscript is not so well-organized and wordy, which makes it difficult to follow. For this reason, I will suggest accepting this manuscript only after re-organizing of the text and considering some major issues listed below.

Major issues:

Section 3.3. focus too much on the technical details in justifying the PMF factors using some independent or internal tracers, which is important but not the scope of this study. Thus, I will only summarize it within few paragraphs, and move some figures and texts into SI. I suggest focusing more on the discussions of each source you retrieved from PMF. The separation between section 3 and 4 and the long text makes the storyline discrete and difficult to follow. In addition, it also makes figures often not the closest to the texts that explain the figures, which makes it difficult to read. Please condense the captions you have in this manuscript, especially for the figures in the SI.

Technical comment:

Have you tried to run the bootstrap analysis to see if you get a rather robust result? If it is just a single PMF run, I would argue you might suffer from rotational ambiguity. How

did you cope with that? Have you ever tried to use factor profiles from "7f_ac_S1_C1" as a-prior information to run PMF but with rather loose constraints?

Detailed comments:

1. Line 73: Add (Lalchandani et al., 2021; Tobler et al., 2020) to your citation.

2. Finger 1: Combine Fig1, Fig.2, and Fig 5 to give a better overview of the chemical composition as well as the data availability

3. Line 129-131: Provide RF, RIEs somewhere, also mention how many times you have calibrated the instrument.

4. Line 124: I will try to mention how many data points were eventually considered and used to run PMF.

5. Line 134-136: Not convinced about how you decide the CE=1 for the Diwali period. Why keep the gradient of 0.8 instead of 0.9, which you obtained from pre- and post-monsoon campaigns?

6. Line 263-264: "The pollution rose suggests that most of the organic mass is from the east and southeast with high peaks (> 140 $\mu$g m-3) originating from the west and north-west." I will add "(Fig. 4(e))" at the end of the sentence. But I cannot draw the same conclusion as you do, I can only see that these high peaks were not originated from east south, northeast, and southwest, the width of the other direction at >140 looks quite similar to me. I will suggest to re-do the plot with more bins of concentration>140 to see if you can see some patterns. Otherwise, I will not make such a statement.

7. Line 264: "Its polar graph also shows some extreme values existing from the south-east." Add "(Fig. S8)" at the end of the sentence to direct readers. Please read through your manuscript again to guide the reader properly.

8. Figure 4: Why there are negative concentration in (b), (d), and (e), what happened to these data points? 1. These plots did not seem to have many fractions for the smallest

concentration bin; 2. I am surprised that you still end up some noisy data points in such a polluted area like India, please explain in more details if you do believe these negative points are real.

9. Figure 6: Please explain what the uncalibrated PAH concentration is in the caption.

10. Line299-300: "The PAH time series in Figure 6 shows similar quantities during the pre-monsoon and monsoon periods and the consistent low levels suggest common sources such as traffic, solid fuel burning or cooking activities." Similar to what? OrgNO? Is the second part of the sentence not finished yet? Or did you mean because the low level of PAH, suggested the sources of PAH are traffic, solid fuel burning, or cooking activities? If so, please add a citation at the end.

11. Line 300-302 "There is however a large increase in PAHs during the post-monsoon period when the burning of the rice crop residue begins," how do you know it started in the post-monsoon, please provide a citation here.

12. Figure 8: label the Diwali period.

13. Line 345: "The PMF results from the AMS unit-mass resolution flux measurements (Ben Langford pers. commun.) show two traffic factors which peak at 9 a.m. and 11 a.m., which supports this." There are some other source apportionment studies in India having HOA factor, I suggest not citing an unpublished manuscript.

14. Figure 9: It is difficult to locate the "Lower panel" when it is a text not a "(b)" in the caption especially when the caption is so long. In addition, for the lower panel graph, I would change the y-axis to mass concentration, and keep the percentage of each factor inside the bar, it makes readers easier to compare the relative concentration of each factor for a different time period. Besides, I think you don't gain extra information here by adding inorganic into this graph when you already have Figure 2.

15. Line 364: "This may suggest atmospheric processing is occurring, for example oxidation reactions." Please explain what oxidation reactions occurred and why it caused

the larger spread of the mass of NHOA in different wind directions. Or did you just simply imply the NHOA was rather fast oxidized that cannot transport in a long-range?

16. Line 437-438: "The polar graph also shows maximum concentrations coincide with a south-easterly direction. The timing of these peaks match with maxima seen in polar graphs for PAH, SFOA, and NHOA, which suggests a common source." Please mention which figure it is since you have two polar graphs and two pollution rose figures.

17. Line 458: "UnSubPAHs" is not defined in the previous text.

18. Line 836-837: "One study indicates that this behaviour results in larger amounts of municipal waste being burnt in the morning compared to the evening (Nagpure et al., 2015)." One could add is that the temperature in India is not low, thus the evening peak is not so pronounced.

19. Line 925: " PCDBs" and "PCDFs" did not mention previously.

Comments for SI:

1. Line numbers are missing

2. The second paragraph on P3: I understand previous studies used the terminology of "background concentration" to define the intercept of the multilinear regression, but I found it was difficult to read directly from Fig. S2-S5 without any other clarification. It is quite miss leading for me, I thought it was an averaged background level in a given time period of one of the tracers, I wonder why it was different over different PMF runs on the same dataset. I will change this terminology or at least descript it in the caption to make the graph easier to read.

3. Last paragraph on P3: use a table instead to explain the name of each solution, it is very difficult to read and understand in the text. Also, in this way, the captions in Fig. S2-S5 could be shortened.

4. Title on P7: "S2. Method for determining the Inorganic-Organic PMF solution"

should be S3

5. Figure S8 on P13 and Figure S10 on P15: add the units for conc. and (a), (b), (c), (d), (e), etc. in the graph. Consider using a color code or anything else to differentiate the wind speed but not with closed circles. It is just not easy to read about the overlapped points, which are true for most of your data points.

6. The first paragraph on P12: "Error! Not a valid bookmark self-reference", please revise.

7. Figure S7 on P19: Could you also label where are the two cremation sites are if you are going to mention them in the text.

---

## Author Comment (AC1) · 20 Apr 2021

**acp-2020-1009: Seasonal analysis of submicron aerosol in Old Delhi using high resolution aerosol mass spectrometry: Chemical characterisation, source apportionment and new marker identification**

**Response to the reviewers' comments**

We thank the reviewer for providing detailed and helpful comments on our manuscript. Below we respond to each reviewer comment in turn (reviewers' comments are shown in italics), indicating the changes we have made.

**Response to Anonymous Referee #1**

The article presents data and findings that are of interest to the scientific community and in a location where air pollution is very high and affects a large number of people. The methods used are sound. The interpretation of the data and the conclusions are well rooted in the data with minimal speculation. The presentation of the data and the results are good, although it could use some more clarity especially in the figures as mentioned below in more detail. The location where the measurements were carried out presents a number of challenges such as high temperatures and high relative humidity that can be really tricky for instruments. I think the authors have generated a great dataset in those challenging conditions. The authors also did an excellent job at interpreting such a complex mixture of sources. The manuscript is of high quality and within the scope of the journal. I recommend the publication after minor revisions.

We thank the reviewer for their very supportive comments on the merits of our study and its presentation.

**Detailed comments:**

1- The manuscript similarly to many manuscripts based on AMS data and PMF, makes extensive use of acronyms. These acronyms are probably very familiar to the authors and to experienced AMS users, however, they tend to be hard to follow for readers less involved with AMS and PMF analysis. I recommend making a list of acronyms to help the reader follow the text.

We agree and thank the reviewer for this suggestion. A list of acronyms has been added in Table S1.

**Table S1 – List of abbreviations**

**Abbreviations**

| IGDTUW              | Indira Gandhi Delhi Technical University for Women         |
|---------------------|------------------------------------------------------------|
| HR-ToF-AMS          | High-resolution time-of-flight aerosol mass spectrometer   |
| PTR-QiTOF           | High-resolution proton transfer reaction mass spectrometer |
| PM1                 | Sub-micron particulate matter                              |
| SOA                 | Secondary organic aerosol                                  |
| VOC                 | Volatile organic compound                                  |
| OrgNO               | Organic nitrogen oxide species                             |
| ВС                  | Black carbon                                               |
| LWC                 | Liquid water content                                       |
| PCDDs               | Polychlorinated dibenzodioxins                             |
| PCDFs               | Polychlorinated dibenzofurans                              |
| $\overline{OS}_{C}$ | Carbon oxidation state                                     |
| PMF                 | Positive Matrix Factorisation                              |
| COA                 | Cooking organic aerosol                                    |
| NHOA                | Nitrogen-containing hydrocarbon-like organic aerosol       |
| SFOA                | Solid fuel organic aerosol                                 |
| НОА                 | Hydrocarbon-like organic aerosol                           |
| SVBBOA              | Semi-volatility biomass burning organic aerosol            |
| LVOOA               | Low-volatility oxygenated organic aerosol                  |
| SVOOA               | Semi-volatility oxygenated organic aerosol                 |
| РАН                 | Polyaromatic hydrocarbons                                  |
| UnSubPAH            | Unsubstituted PAH                                          |
| МРАН                | Methyl-substituted PAH                                     |
| ОРАН                | Oxidised PAH                                               |
| NOPAH               | Nitrogen-oxygen substituted PAH                            |
| АРАН                | Amino PAH                                                  |
| VK                  | Van Krevelen                                               |

2- Figure 1. I suggest adding the time series of the standard AMS species (NO3, SO4, NH4, Chl, and Org) and BC. This will give the reader a good bird's eye view of the dataset, maybe merging it with some version of Figure 5.

Figures 1, 2 and 5 have been combined to produce a new summary figure (Figure 1). Please note that the collection efficiency during the Diwali Period (05 to 14/11/18) has been changed to a lower CE of 0.8 in order to obtain a PM1 vs. PM2.5 gradient (~0.9) that is similar to the other measurement periods. This change resulted from a suggestion made by the second reviewer. The concentration results have therefore slightly changed in many averages and statistics, and all the relevant figures and tables have been updated accordingly.

---

## Author Comment (AC2) · 20 Apr 2021

**acp-2020-1009: Seasonal analysis of submicron aerosol in Old Delhi using high resolution aerosol mass spectrometry: Chemical characterisation, source apportionment and new marker identification**

**Response to the reviewers' comments**

We thank the reviewer for providing detailed and helpful comments on our manuscript. Below we respond to each reviewer comment in turn (reviewers' comments are shown in italics), indicating the changes we have made.

Response to Anonymous Referee #2

*General Comments: The manuscript by Cash et al. describes a chemical characterization and source apportionment work using high resolution aerosol mass spectrometry (HR-AMS) data collected in one of the most polluted and populated areas in Old Delhi, India in 2018. It covered 3 time periods: pre-monsoon (around 1 month), monsoon (around 15 days), and post-monsoon (around 6 weeks). The study identified two traffic-related factors (NHOA and HOA), two burning-related factors (SFOA and SVBBOA), a cooking factor (COA), and two SOA factors (LVOOA and SVOOA) using positive matrix factorization (PMF). In the end, the Authors suggest developing air quality policies by mitigating emissions from crop residue burning, open waste burning, and traffic activities. Cash et al. were able to reveal the mystery of high chloride concentration using HR-AMS, which has strong links with the burning sources, especially in the post-monsoon period. In addition, with the rather well-resolved PMF solution, it is certainly a valuable reference for AMS users in the future (e.g., the mass spectra of NHOA, SFOA, SVBBOA). Collecting and analyzing such a dataset from a high temperature, high relative humidity environment remains challenging, but the authors were able to justify their results using both independent and internal tracers. The multilinear regression analysis is a useful tool for this dataset considering there is no reference HR mass spectrum available for a unique environment, like old Delhi. I believe the contribution of this work is significant and meets the scope of the ACP. Overall, the English are perfect, but the whole manuscript is not so well-organized and wordy, which makes it difficult to follow. For this reason, I will suggest accepting this manuscript only after re-organizing of the text and considering some major issues listed below.*

*Major issues: Section 3.3. focus too much on the technical details in justifying the PMF factors using some independent or internal tracers, which is important but not the scope of this study. Thus, I will only summarize it within few paragraphs, and move some figures and texts into SI. I suggest focusing more on the discussions of each source you retrieved from PMF. The separation between section 3 and 4 and the long text makes the storyline discrete and difficult to follow. In addition, it also makes figures often not the closest to the texts that explain the*

*figures, which makes it difficult to read. Please condense the captions you have in this manuscript, especially for the figures in the SI.*

We have removed parts of the text in the results section that on further consideration we felt were less relevant and we have moved other parts to the SI. Some figures (Figures 12(a) and 13) that are not key to the storyline have also been moved into the SI (Now Figures S13 and S14). We thank the reviewer for these suggestions as they have improved the transition from the results to the discussion section.

In response to the reviewer's comment on this matter we have altered and reduced the majority of the figure captions. However, we are strongly of the opinion that they should retain sufficient information to permit figures to be understandable to the reader without reading information from the main text.

*Technical comment: Have you tried to run the bootstrap analysis to see if you get a rather robust result? If it is just a single PMF run, I would argue you might suffer from rotational ambiguity. How did you cope with that? Have you ever tried to use factor profiles from "7f_ac_S1_C1" as a-prior information to run PMF but with rather loose constraints?*

We did conduct bootstrapping analysis but, the results showed variability in the solutions. The variation calculated through bootstrapping analysis includes mainly random error with partial contribution from rotational ambiguity (Brown et al., 2015). Little rotational ambiguity was found when using FPEAKS, which means the variation found from bootstrapping is most likely related to random error. However, the solutions were found to contain a high degree of correlation between factors meaning that bootstrapping analysis may be incorrect or ambiguous for this dataset (Ulbrich, 2011).

The boostrapping resampling method could be potentially altered in order to solve this issue. The resampling method used by the PET tool (used in this study) resamples by randomly replacing subsets of original rows (mass spectra) with other rows from the original matrix (Ulbrich, 2011). The EPA PMF software however goes one step further and uses a resampling method called block bootstrapping where a new dataset is created from randomly selecting non-overlapping time periods or 'blocks' (i.e. of length 3-days) (Paatero et al., 2014). This is likely a better method for this dataset and it may help to reduce the effects of serial correlation (or correlation between factors). It is however unlikely to account for large variations in factors between different time periods. Factors such as SVBBOA, for example, increase from 0 to 40 μg m$^{-3}$ going from pre-monsoon to post-monsoon. After resampling, 'blocks' from the pre-monsoon will be placed next to 'blocks' from the post-monsoon and this will create a large amount of variation that is not due to model error or solution instability. Previous studies have described similar issues and have used their own resampling method (Hemann et al., 2009).

It may be possible to apply bootstrapping analysis on a seasonal basis to counteract these issues, but this is beyond the scope of this study. There is also the time series dependence on the boundary layer which is unlikely to be accounted for using standard resampling methods. Additionally, it is computationally demanding and software, such as EPA PMF, do not support the analysis of a $2.39673 \times 10^7$ data point matrix.

Using SEED analysis, we determined that there is variance in the solution and to deal with this potential ambiguity we applied multilinear regression analysis on possible SEEDS which is a well establish method (Allan et al., 2010; Young et al., 2015; Elser et al., 2016; Reyes-Villegas et al., 2016). This method deals with time series dependencies as the external tracers are similarly affected by the same meteorological effects such as the boundary layer.

We have not tried to use the profiles as a priori information to run in ME-2 but unless we constrain each factor differently, we fail to see how this would change the result from using variations in the SEED. If factors were to be constrained differently to one another, there are no previous high-resolution studies in Delhi on which to base this. We are therefore strongly of the opinion that applying no constraints to novel PMF profiles provides a more objective result.

*Detailed comments:*

*1. Line 73: Add (Lalchandani et al., 2021; Tobler et al., 2020) to your citation.*

These citations have been added: "A growing number of studies in Delhi, and other locations in India, have reported large concentrations of chloride, especially during the morning hours at ~7-9 a.m. (Sudheer et al., 2014; Chakraborty et al., 2018; Gani et al., 2019; Acharja et al., 2020; Reyes-Villegas et al., 2020; Tobler et al., 2020; Lalchandani et al., 2021)."

*2. Finger 1: Combine Fig1, Fig.2, and Fig 5 to give a better overview of the chemical composition as well as the data availability.*

We thank the reviewer for this suggestion and have combined the three figures to create a better summary of the data (new Figure 1).

[Figure]

**Figure 1 – First Panel: Average relative contributions of chloride, ammonium, nitrate, sulphate, organic aerosol and black carbon to the total PM₁ mass loadings in the pre-monsoon, monsoon and post-monsoon periods. The average concentrations of each species are shown to the right of each bar (see Table S5 for values and statistics). Second panel: Gant chart showing the measurement periods where the red region shows the Diwali festival and the green region shows when the inlet was moved to a 30 m tower. Third panel: time series of the relative humidity and the temperature for the three measurement periods. Fourth panel: time series of the wind speed with arrows showing wind direction. Fifth panel: time series of stacked concentrations of aerosol species showing total PM₁.**

*3. Line 129-131: Provide RF, RIEs somewhere, also mention how many times you have calibrated the instrument.*

Details on the calibration results are summarised in Table S2 and the number of times the HR-AMS was calibrated have been added to the text: "The HR-TOF-AMS was calibrated fortnightly over the three campaigns (11 calibrations in total) for its ionisation efficiency of nitrate (IE) and the relative ionisation efficiency (RIE)..."

**Table S2 – Relative ionisation efficiencies (RIE), ionisation efficiencies (IE) and collection efficiencies (CE)**

| Season | IE | RIE NH$_4^+$ | RIE SO$_4^{2-}$ | RIE Cl$^-$ | CE |
|---|---|---|---|---|---|
| Pre-monsoon | 2.92E-07 | 4 | 1.45 | 2.07 | 0.5 |
| Monsoon | 2.92E-07 | 4 | 1.45 | 2.07 | 0.5 |
| Post-monsoon preflux period (11/10/18 - 03/11/18) | 2.89E-07 | 4 | 1.45 | 2.07 | 0.5 |
| Post-monsoon Diwali period (05/11/18 - 14/11/18) | 3.14E-07 | 4 | 1.45 | 1.05 | 0.8 |
| Post-monsoon post-Diwali (14/11/18 - 23/11/18) | 3.14E-07 | 4 | 1.45 | 1.05 | 0.5 |

*4. Line 124: I will try to mention how many data points were eventually considered and used to run PMF.*
The number of data points within the organic matrix have been added: "The primary PMF analysis was conducted on the organic matrix ($2.39673 \times 10^7$ data points)..."

*5. Line 134-136: Not convinced about how you decide the CE=1 for the Diwali period. Why keep the gradient of 0.8 instead of 0.9, which you obtained from pre- and post monsoon campaigns?*
The collection efficiency for the Diwali period has been changed to CE=0.8 to maintain the gradient of 0.9. This has resulted in a slight change in many averages and statistics, and all the relevant figures and tables have been updated accordingly.

*6. Line 263-264: "The pollution rose suggests that most of the organic mass is from the east and southeast with high peaks (> 140 µg m-3) originating from the west and northwest." I will add "(Fig. 4(e))" at the end of the sentence. But I cannot draw the same conclusion as you do, I can only see that these high peaks were not originated from east south, northeast, and southwest, the width of the other direction at >140 looks quite similar to me. I will suggest to re-do the plot with more bins of concentration>140 to see if you can see some patterns. Otherwise, I will not make such a statement.*
We have now provided a new version of the pollution rose figure for the organics (new Figure 3). There are now more meaningful breaks in the concentration bins and the sizing of the wind vectors have been changed to show the contribution of a particular wind direction to the mean concentration (rather than to the frequency of counts, as before). The statement has been altered to better reflect the data shown in the newly edited figure: "The pollution rose and polar graph for organic aerosol are highly spread. The pollution rose suggests that there is a slight increase in the organic mass when originating from the east and south east but this is closely followed by contributions from the west and northwest (Figure 3e)."

[Figure]

**Figure 3 – Pollution roses for (a) chloride, (b) ammonium, (c) nitrate, (d) sulphate and (e) organic aerosol, along with (f) a wind rose plot for all measurement periods combined. The pollution roses show 30º wind vectors and their size is proportional to the percentage contribution to the mean concentration. The vectors are divided into concentration bins based on the colour scale in the legend.**

*7. Line 264: "Its polar graph also shows some extreme values existing from the southeast." Add "(Fig. S8)" at the end of the sentence to direct readers. Please read through your manuscript again to guide the reader properly.*

We have added a reference to Figure S8: "Its polar graph also shows some extreme values existing from the south east (Figure S8)."

*8. Figure 4: Why there are negative concentration in (b), (d), and (e), what happened to these data points? 1. These plots did not seem to have many fractions for the smallest concentration bin; 2. I am surprised that you still end up some noisy data points in such a polluted area like India, please explain in more details if you do believe these negative points are real.*

We agree that these are incorrect and we found an error in the concentration scale which has been corrected. The pollution rose has also been altered to better reflect the direction of mass associated with wind direction. Previously the graphs were showing bin sizes based on frequency of counts in a particular wind direction but now they are sized to show the percentage contribution to the concentration mean. This creates a clearer picture of wind directional preferences (see new Figure 3).

*9. Figure 6: Please explain what the uncalibrated PAH concentration is in the caption.*

We have provided an explanation in the text when the PAH time series figure is first introduced: " The time series of PAH concentrations in Figure 4 are uncalibrated concentrations and further lab work (such as the work carried out in Herring et al. (2015)) is required to establish absolute concentrations which goes beyond the scope of this study." Please note that Figure 6 is now Figure 4.

*10. Line299-300: "The PAH time series in Figure 6 shows similar quantities during the pre-monsoon and monsoon periods and the consistent low levels suggest common sources such as traffic, solid fuel burning or*

*cooking activities." Similar to what? OrgNO? Is the second part of the sentence not finished yet? Or did you mean because the low level of PAH, suggested the sources of PAH are traffic, solid fuel burning, or cooking activities? If so, please add a citation at the end.*

We apologise that the English was not quite correct here and we have altered the text to explain that PAHs are similar in quantity during the monsoon and pre-monsoon. Due to the fairly consistent level of PAHs, we suggest that the dominant sources during these periods are also consistent emitters such as traffic, solid fuel burning and cooking activities: "The data does, however, show the relative change in PAHs is small between the pre-monsoon and monsoon periods suggesting consistent sources are responsible such as traffic, solid fuel burning or cooking activities."

*11. Line 300-302 "There is however a large increase in PAHs during the post-monsoon period when the burning of the rice crop residue begins," how do you know it started in the post-monsoon, please provide a citation here.*

A citation has been added to the statement (Kulkarni et al., 2020).

*12. Figure 8: label the Diwali period.*

The Diwali period has been labelled using a light blue shading in (see new Figure 6).

[Figure]

**Figure 6 - Time series for each factor where the x-axis is broken to show each measurement period. Regions shaded in grey are night hours and the Diwali period is shaded in light blue. The time series of the normalised concentrations are also shown in the bottom section of the graph.**

*13. Line 345: "The PMF results from the AMS unit-mass resolution flux measurements (Ben Langford pers. commun.) show two traffic factors which peak at 9 a.m. and 11 a.m., which supports this." There are some other source apportionment studies in India having HOA factor, I suggest not citing an unpublished manuscript.*

We have updated the citation to a European Aerosol Conference abstract which summarised the results (Di Marco et al., 2019).

*14. Figure 9: It is difficult to locate the "Lower panel" when it is a text not a "(b)" in the caption especially when the caption is so long. In addition, for the lower panel graph, I would change the y-axis to mass concentration, and keep the percentage of each factor inside the bar, it makes readers easier to compare the relative concentration of each factor for a different time period. Besides, I think you don't gain extra information here by adding inorganic into this graph when you already have Figure 2.*

The upper and lower panels on this figure now have additional (a) and (b) labels (see Figure 7). If the y-axis is altered to mass concentration in Figure 7(b), it becomes particularly hard to see changes between the monsoon and pre-monsoon as the post-monsoon is 3 times larger in mass and dominates the graph. The inorganics are added to show the total $PM_1$ which makes it easier to view the fraction of total $PM_1$ mass that is due to each factor. This gives factor mass more meaning and we are reluctant to remove the inorganics for this reason.

[Figure]

**Figure 7 – Correlation matrix (a) between the AMS OA factors (bold), internal tracers and external tracers for the combined dataset (all seasons). The correlation coefficients are ordered using hierarchical cluster analysis. The relative contribution of OA factors, BC and inorganic species to total PM$_1$ for each period (b) where the total average PM$_1$ is shown using right curly brackets (see Table S5 for values and statistics). The three ions [C$_6$H$_{10}$O]$^+$, [C$_8$H$_8$O$_2$]$^+$ and [C$_4$H$_4$O]$^+$ are, respectively: an organic acid fragment used for COA determination, a fragment of dibenzodioxin called benzodioxan and, furan a fragment of dibenzofuran. The abbreviations of VOC species are: Ace = acetonitrile, Ben = benzene, Tol = Toluene and EBen = Ethylbenzene.**

*15. Line 364: "This may suggest atmospheric processing is occurring, for example oxidation reactions." Please explain what oxidation reactions occurred and why it caused the larger spread of the mass of NHOA in different wind directions. Or did you just simply imply the NHOA was rather fast oxidized that cannot transport in a long-range?*

We have added the following text to provide a clearer explanation: "A larger spread for NHOA when compared to HOA could suggest that atmospheric processing is occurring as both have similar traffic origins. NHOA could therefore be a result of aged or oxidised traffic emissions allowing it to be transported further distances and causing a mixture of wind directional preferences."

*16. Line 437-438: "The polar graph also shows maximum concentrations coincide with a south-easterly direction. The timing of these peaks match with maxima seen in polar graphs for PAH, SFOA, and NHOA, which suggests a common source." Please mention which figure it is since you have two polar graphs and two pollution rose figures.*

References to figures have been added within the text: "The polar graph also shows maximum concentrations coincide with a south-easterly direction (Figure S10h). The timing of these peaks match with maxima seen in polar graphs for PAH, SFOA and NHOA (Figure S10), which suggests a common source."

*17. Line 458: "UnSubPAHs" is not defined in the previous text.*

This has now been defined within the text: "The PAH composition of COA is mainly unsubstituted PAHs (UnSubPAHs) and also contributes the largest…"

*18. Line 836-837: "One study indicates that this behaviour results in larger amounts of municipal waste being burnt in the morning compared to the evening (Nagpure et al., 2015)." One could add is that the temperature in India is not low, thus the evening peak is not so pronounced.*

We have added this observation to the following text: "One study indicates that this behaviour results in larger amounts of municipal waste being burnt in the morning compared to the evening (Nagpure et al., 2015). Additionally, the temperature in Delhi does not drop until later in the evening making it less likely that residents burn refuse to stay warm in the earlier parts of the evening."

*19. Line 925: " PCDBs" and "PCDFs" did not mention previously.*

These have now been defined within the text: "We introduce novel AMS measured tracers which show the presence of polychlorinated dibenzo-furans (PCDF) and -dioxins (PCDD). This offers a new way to associate sources to plastic or Cl-rich fuel burning."

*Comments for SI:*
*1. Line numbers are missing*

Line numbers have been added.

*2. The second paragraph on P3: I understand previous studies used the terminology of "background concentration" to define the intercept of the multilinear regression, but I found it was difficult to read directly*

*from Fig. S2-S5 without any other clarification. It is quite miss leading for me, I thought it was an averaged background level in a given time period of one of the tracers, I wonder why it was different over different PMF runs on the same dataset. I will change this terminology or at least descript it in the caption to make the graph easier to read.*

The description of the background concentration being estimated using the intercept of the linear regression is included in the figure captions (e.g. see Figure S2). This description is also included in the text within the subsection 'Step 2' of section S1. "The coefficients are valuable ways of verifying the solution fit where *A* is an indication of the background concentration of the tracer."

[Figure]

**Figure S1 – Trilinear regression analysis results for the PMF solutions taken from the all-periods-combined case. Results are shown for the fit using (a) CO, (b) NOₓ and (c) BC as external tracers. Gradient contributions for factors SFOA, COA and HOA are shown alongside the background concentration of the tracer (black) which is estimated using the intercept of the linear regression. The chi-square value (red markers), the Q/Qₑₓₚ (blue markers) and the chosen final solution (labelled with a blue arrow) are also shown below.**

*3. Last paragraph on P3: use a table instead to explain the name of each solution, it is very difficult to read and understand in the text. Also, in this way, the captions in Fig.S2-S5 could be shortened.*

See Table S3 for a description of solution labels. The description of solution labels within Figure captions has also been removed.

**Table S3 – A description of labels used to define a solution.**

| Label (Xf_ac_<120*m/z*_SX_CX) | Description |
|---|---|
| Xf, where X=1,2,3,n | The number of factors |
| ac | If present, this indicates it is resolved from the all-combined analysis case (all periods analysed in one PMF analysis). Otherwise, the solution is limited to a specific measurement period |
| <120mz | If present, this indicates the solution is limited to include ions up to *m/z* 120. Otherwise, ions up to *m/z* 385 are included |
| SX, where X=1,2,3,n | The SEED number |
| CX, where X=1,2,3,n | Indicates a specific combination of factors used for the *SFOA* variable in Eq. (S1) an SFOA factor time series alone is used for the *SFOA* variable |
| C1 | a combined time series of an SFOA and an SVBBOA factor |
| C2 | a combined time series of two SFOA factors (for solutions that produce two SFOA |
| C3 | factors) |

*4. Title on P7: "S2. Method for determining the Inorganic-Organic PMF solution" should be S3*

The section heading has been changed.

*5. Figure S8 on P13 and Figure S10 on P15: add the units for conc. and (a), (b), (c), (d), (e), etc. in the graph. Consider using a color code or anything else to differentiate the wind speed but not with closed circles. It is just not easy to read about the overlapped points, which are true for most of your data points.*

Units and (a), (b), (c), etc. labels have been added to the figure (see Figure S8 and S10). When using a colour scale, the same issue occurs with overlapping of data points. The spread of the data is better presented in the revised pollution roses in the main text (new Figure 3 and 9). The polar plots are therefore only present to emphasise the maximum concentrations and their origin. This is highlighted in the SI text: "The polar plots offer additional information about the highest values which are seen more clearly in Figure S8 and show the direction of possible high source contributors."

[Figure]

**Figure S8 – Polar graphs showing the concentrations (in µg m⁻³) by wind direction for chloride, ammonium, nitrate, sulphate and organics for all measurement periods combined. Each point represents a 5-minute measurement. Open symbols show concentrations for winds speeds <1 m s⁻¹ and closed symbols for wind speeds >1 m s⁻¹.**

[Figure]

**Figure S10 – Polar graphs showing the concentrations (in µg m⁻³) by wind direction for the factor solutions: (a) COA, (b) NHOA, (c) SFOA, (d) HOA, (e) SVBBOA, (f) LVOOA and (g) SVOOA. (h) PAH polar graph shows uncalibrated concentrations (a.u.) by wind direction. Each point represents a 30-minute average measurement. Open symbols show concentrations for winds speeds <1 m s⁻¹ and closed symbols for wind speeds >1 m s⁻¹.**

[Figure]

**Figure 9 - Pollution roses for each factor and uncalibrated PAH concentrations along with a wind rose. The pollution roses show 30° wind vectors and their size is proportional to the percentage contribution to the mean concentration. The vectors are divided into concentration bins based on the colour scale in the legend.**

*6. The first paragraph on P12: "Error! Not a valid bookmark self-reference", please revise.*

Correction made.

*7. Figure S7 on P19: Could you also label where are the two cremation sites are if you are going to mention them in the text.*

The cremation site has been labelled on Figure S16. Upon inspection of the second infant cremation site, we are unsure whether this site would truly contribute to PM measured at IGDTUW. This site is also particularly small in comparison to the very large cremation site nearby. We have therefore chosen to remove references to the infant cremation site.

[Figure]

**Figure S16 – Monitoring site map on (a) small and (b) large scale. The red circles show the monitoring site location and a nearby large cremation site in (a).**

References:

Acharja, P., Ali, K., Trivedi, D. K., Safai, P. D., Ghude, S., Prabhakaran, T. and Rajeevan, M.: Characterization of atmospheric trace gases and water soluble inorganic chemical ions of PM1 and PM2.5 at Indira Gandhi International Airport, New Delhi during 2017–18 winter, Sci. Total Environ., 729, 138800, doi:10.1016/j.scitotenv.2020.138800, 2020.

Allan, J. D., Williams, P. I., Morgan, W. T., Martin, C. L., Flynn, M. J., Lee, J., Nemitz, E., Phillips, G. J., Gallagher, M. W. and Coe, H.: Contributions from transport, solid fuel burning and cooking to primary organic

aerosols in two UK cities, Atmos. Chem. Phys., 10(2), 647–668, doi:10.5194/acp-10-647-2010, 2010.

Brown, S. G., Eberly, S., Paatero, P. and Norris, G. A.: Methods for estimating uncertainty in PMF solutions: Examples with ambient air and water quality data and guidance on reporting PMF results, Sci. Total Environ., 518–519, 626–635, doi:10.1016/j.scitotenv.2015.01.022, 2015.

Chakraborty, A., Mandariya, A. K., Chakraborti, R., Gupta, T. and Tripathi, S. N.: Realtime chemical characterization of post monsoon organic aerosols in a polluted urban city: Sources, composition, and comparison with other seasons, Environ. Pollut., 232, 310–321, doi:10.1016/j.envpol.2017.09.079, 2018.

Elser, M., Huang, R. J., Wolf, R., Slowik, J. G., Wang, Q., Canonaco, F., Li, G., Bozzetti, C., Daellenbach, K. R., Huang, Y., Zhang, R., Li, Z., Cao, J., Baltensperger, U., El-Haddad, I. and André, P.: New insights into PM2.5 chemical composition and sources in two major cities in China during extreme haze events using aerosol mass spectrometry, Atmos. Chem. Phys., 16(5), 3207–3225, doi:10.5194/acp-16-3207-2016, 2016.

Gani, S., Bhandari, S., Seraj, S., Wang, D. S., Patel, K., Soni, P., Arub, Z., Habib, G., Hildebrandt Ruiz, L. and Apte, J. S.: Submicron aerosol composition in the world's most polluted megacity: The Delhi Aerosol Supersite study, Atmos. Chem. Phys., 19(10), 6843–6859, doi:10.5194/acp-19-6843-2019, 2019.

Hemann, J. G., Brinkman, G. L., Dutton, S. J., Hannigan, M. P., Milford, J. B. and Miller, S. L.: Assessing positive matrix factorization model fit: A new method to estimate uncertainty and bias in factor contributions at the measurement time scale, Atmos. Chem. Phys., 9(2), 497–513, doi:10.5194/acp-9-497-2009, 2009.

Herring, C. L., Faiola, C. L., Massoli, P., Sueper, D., Erickson, M. H., McDonald, J. D., Simpson, C. D., Yost, M. G., Jobson, B. T. and Van Reken, T. M.: New Methodology for Quantifying Polycyclic Aromatic Hydrocarbons (PAHs) Using High-Resolution Aerosol Mass Spectrometry, Aerosol Sci. Technol., 49(11), 1131–1148, doi:10.1080/02786826.2015.1101050, 2015.

Kulkarni, S. H., Ghude, S. D., Jena, C., Karumuri, R. K., Sinha, B., Sinha, V., Kumar, R., Soni, V. K. and Khare, M.: How Much Does Large-Scale Crop Residue Burning Affect the Air Quality in Delhi?, Environ. Sci. Technol., 54(8), 4790–4799, doi:10.1021/acs.est.0c00329, 2020.

Lalchandani, V., Kumar, V., Tobler, A., M. Thamban, N., Mishra, S., Slowik, J. G., Bhattu, D., Rai, P., Satish, R., Ganguly, D., Tiwari, S., Rastogi, N., Tiwari, S., Močnik, G., Prévôt, A. S. H. and Tripathi, S. N.: Real-time characterization and source apportionment of fine particulate matter in the Delhi megacity area during late winter, Sci. Total Environ., 770, 145324, doi:10.1016/j.scitotenv.2021.145324, 2021.

Di Marco, C. D., Langford, B., Cash, J. M., Mullinger, N., Helfter, C. and Nemitz, E.: Source apportionment analysis applied to aerosol eddy-covariance fluxes in Delhi, European Aerosol Conference. [online] Available from: https://www.costcolossal.eu/specialsessioneac2019/, 2019.

Nagpure, A. S., Ramaswami, A. and Russell, A.: Characterizing the Spatial and Temporal Patterns of Open Burning of Municipal Solid Waste (MSW) in Indian Cities, Environ. Sci. Technol., 49(21), 12911–12912, doi:10.1021/acs.est.5b03243, 2015.

Paatero, P., Eberly, S., Brown, S. G. and Norris, G. A.: Methods for estimating uncertainty in factor analytic solutions, Atmos. Meas. Tech., 7(3), 781–797, doi:10.5194/amt-7-781-2014, 2014.

Reyes-Villegas, E., Green, D. C., Priestman, M., Canonaco, F., Coe, H., Prévôt, A. S. H. and Allan, J. D.: Organic aerosol source apportionment in London 2013 with ME-2: Exploring the solution space with annual and seasonal analysis, Atmos. Chem. Phys., 16(24), 15545–15559, doi:10.5194/acp-16-15545-2016, 2016.

Reyes-Villegas, E., Panda, U., Darbyshire, E., Cash, J. M., Joshi, R., Langford, B., Di Marco, C. F., Mullinger,

N., Acton, W. J. F., Drysdale, W., Nemitz, E., Flynn, M., Voliotis, A., McFiggans, G., Coe, H., Lee, J., Hewitt, C. N., Heal, M. R., Gunthe, S. S., Shivani, Gadi, R., Singh, S., Soni, V. and Allan, J. D.: PM1 composition and source apportionment at two sites in Delhi, India across multiple seasons, Atmos. Chem. Phys. Discuss., 2020, 1–19, doi:10.5194/acp-2020-894, 2020.

Sudheer, A. K., Rengarajan, R., Deka, D., Bhushan, R., Singh, S. K. and Aslam, M. Y.: Diurnal and seasonal characteristics of aerosol ionic constituents over an urban location in Western India: Secondary aerosol formation and meteorological influence, Aerosol Air Qual. Res., 14(6), 1701–1713, doi:10.4209/aaqr.2013.09.0288, 2014.

Tobler, A., Bhattu, D., Canonaco, F., Lalchandani, V., Shukla, A., Thamban, N. M., Mishra, S., Srivastava, A. K., Bisht, D. S., Tiwari, S., Singh, S., Močnik, G., Baltensperger, U., Tripathi, S. N., Slowik, J. G. and Prévôt, A. S. H.: Chemical characterization of PM2.5 and source apportionment of organic aerosol in New Delhi, India, Sci. Total Environ., 745, 140924, doi:10.1016/j.scitotenv.2020.140924, 2020.

Ulbrich, I. M.: Characterization of Positive Matrix Factorization Methods and Their Application to Ambient Aerosol Mass Spectra, Chem. Biochem. Grad. Theses Diss., Paper 35 [online] Available from: https://core.ac.uk/download/pdf/54848806.pdf, 2011.

Young, D. E., Allan, J. D., Williams, P. I., Green, D. C., Harrison, R. M., Yin, J., Flynn, M. J., Gallagher, M. W. and Coe, H.: Investigating a two-component model of solid fuel organic aerosol in London: Processes, PM1 contributions, and seasonality, Atmos. Chem. Phys., 15(5), 2429–2443, doi:10.5194/acp-15-2429-2015, 2015.

---

## Author Comment (AC3) · 21 Apr 2021

acp-2020-1009: Seasonal analysis of submicron aerosol in Old Delhi using high resolution aerosol mass spectrometry: Chemical characterisation, source apportionment and new marker identification

**Response to the reviewers' comments**

We thank the reviewer for providing detailed and helpful comments on our manuscript. Below we respond to each reviewer comment in turn (reviewers' comments are shown in italics), indicating the changes we have made.

Response to Anonymous Referee #1

*The article presents data and findings that are of interest to the scientific community and in a location where air pollution is very high and affects a large number of people. The methods used are sound. The interpretation of the data and the conclusions are well rooted in the data with minimal speculation. The presentation of the data and the results are good, although it could use some more clarity especially in the figures as mentioned below in more detail. The location where the measurements were carried out presents a number of challenges such as high temperatures and high relative humidity that can be really tricky for instruments. I think the authors have generated a great dataset in those challenging conditions. The authors also did an excellent job at interpreting such a complex mixture of sources. The manuscript is of high quality and within the scope of the journal. I recommend the publication after minor revisions.*

We thank the reviewer for their very supportive comments on the merits of our study and its presentation.

*Detailed comments:*

*1- The manuscript similarly to many manuscripts based on AMS data and PMF, makes extensive use of acronyms. These acronyms are probably very familiar to the authors and to experienced AMS users, however, they tend to be hard to follow for readers less involved with AMS and PMF analysis. I recommend making a list of acronyms to help the reader follow the text.*

We agree and thank the reviewer for this suggestion. A list of acronyms has been added in Table S1.

**Table S1 – List of abbreviations**

Abbreviations

| | |
|---|---|
| IGDTUW | Indira Gandhi Delhi Technical University for Women |
| HR-ToF-AMS | High-resolution time-of-flight aerosol mass spectrometer |
| PTR-QiTOF | High-resolution proton transfer reaction mass spectrometer |
| $PM_1$ | Sub-micron particulate matter |
| SOA | Secondary organic aerosol |
| VOC | Volatile organic compound |
| OrgNO | Organic nitrogen oxide species |
| BC | Black carbon |
| LWC | Liquid water content |
| PCDDs | Polychlorinated dibenzodioxins |
| PCDFs | Polychlorinated dibenzofurans |
| $\overline{OS}_C$ | Carbon oxidation state |
| PMF | Positive Matrix Factorisation |
| COA | Cooking organic aerosol |
| NHOA | Nitrogen-containing hydrocarbon-like organic aerosol |
| SFOA | Solid fuel organic aerosol |
| HOA | Hydrocarbon-like organic aerosol |
| SVBBOA | Semi-volatility biomass burning organic aerosol |
| LVOOA | Low-volatility oxygenated organic aerosol |
| SVOOA | Semi-volatility oxygenated organic aerosol |
| PAH | Polyaromatic hydrocarbons |
| UnSubPAH | Unsubstituted PAH |
| MPAH | Methyl-substituted PAH |
| OPAH | Oxidised PAH |
| NOPAH | Nitrogen-oxygen substituted PAH |
| APAH | Amino PAH |
| VK | Van Krevelen |

*2- Figure 1. I suggest adding the time series of the standard AMS species (NO3, SO4, NH4, Chl, and Org) and BC. This will give the reader a good bird's eye view of the dataset, maybe merging it with some version of Figure 5.*

Figures 1, 2 and 5 have been combined to produce a new summary figure (Figure 1). Please note that the collection efficiency during the Diwali Period (05 to 14/11/18) has been changed to a lower CE of 0.8 in order to obtain a $PM_1$ vs. $PM_{2.5}$ gradient (~0.9) that is similar to the other measurement periods. This change resulted from a suggestion made by the second reviewer. The concentration results have therefore slightly changed in many averages and statistics, and all the relevant figures and tables have been updated accordingly.

[Figure]

**Figure 1 – First Panel: Average relative contributions of chloride, ammonium, nitrate, sulphate, organic aerosol and black carbon to the total PM₁ mass loadings in the pre-monsoon, monsoon and post-monsoon periods. The average concentrations of each species are shown to the right of each bar (see Table S5 for values and statistics). Second panel: Gant chart showing the measurement periods where the red region shows the Diwali festival and the green region shows when the inlet was moved to a 30 m tower. Third panel: time series of the relative humidity and the temperature for the three measurement periods. Fourth panel: time series of the wind speed with arrows showing wind direction. Fifth panel: time series of stacked concentrations of aerosol species showing total PM₁.**

*3- Figure 3. I suggest adding dark and light hours with a shaded area for the transition/changing light conditions over the measurement period to help the reader form a picture of the data presented.*

Shading for dark and light hours has been added to figures 2 and 8.

[Figure]

**Figure 2 – Median diurnal cycles for aerosol chemical species and for BC, CO and NO$_x$ during the (a) pre-monsoon, (b) monsoon and (c) post-monsoon periods. The median concentration is represented by the thick line and the interquartile range is represented by the shading. Regions shaded in grey are night hours. Data for CO and NO$_x$ are not available for the monsoon period.**

*4- A lot of figures have tiny labels that are really hard to read especially once the manuscript is printed. In Figure 4 the percentage numbers in each panel are very small I suggest using fewer vales and a larger font. Also, "mean" and "calm" (panels ae), as well as the Wind Speed values in panel (f) are almost not readable in the printed version.*

The wind and pollution roses (now presented in Figures 3 and 9) have been increased in size and altered to show wind vectors that are sized based on the contribution to the mean (rather than sized based on frequency of counts). We hope this creates a clearer picture for observing the wind directional relationships within the data. The font size of all labels have also been increased.

[Figure]

**Figure 3 – Pollution roses for (a) chloride, (b) ammonium, (c) nitrate, (d) sulphate and (e) organic aerosol, along with (f) a wind rose plot for all measurement periods combined. The pollution roses show 30º wind vectors and their size is proportional to the percentage contribution to the mean concentration. The vectors are divided into concentration bins based on the colour scale in the legend.**

*5- Figure 5. I suggest merging it with figure 1 as mentioned in comment 2*

This has been done – see previous response.

*6- Figure 6. Dates are very small, please increase the font.*

The font size has been increased on this figure (now Figure 4).

[Figure]

**Figure 4 – Upper panel: time series of $NO_2^+/NO^+$ ratio in the three measurement periods. Lower panel: time series of polyaromatic hydrocarbon (PAH) uncalibrated concentrations and organic nitrogen oxide species (OrgNO$_{mass}$) concentrations.**

*7- Figure 7 the y axis could be harmonized. It's ok to keep a different scale but I suggest keeping the same number of ticks.*

The y-axis has been harmonized on this figure (now Figure 5).

[Figure]

**Figure 5 – Organic-only PMF solution with elemental ratios shown for each factor in the left-hand corner of each spectrum. The mass spectra on the left show *m/z* 12-100 on a linear scale, while the spectra to the right show *m/z* 100-320 on a logarithmic scale. The peaks are coloured based on the chemical families shown in the legend.**

*8- Figure 10. the y axis labels for the O:C, H:C and N:C ratios panel are too small. Please reduce the number of ticks, decide how many to put there (3?) and maybe increase the font.*

All axes labels have been increased in font size and the number of ticks have been reduced on this figure (now Figure 8).

[Figure]

**Figure 8 – Median diurnal cycles of the factor solutions for the three measurement periods (interquartile range indicated by the shading) along with the elemental ratios. Regions shaded in grey are night hours.**

*9- Figure 11. "mean" and "calm" almost not readable*

The font size has been increased on this figure (now Figure 9).

[Figure]

**Figure 9 - Pollution roses for each factor and uncalibrated PAH concentrations along with a wind rose. The pollution roses show 30° wind vectors and their size is proportional to the percentage contribution to the mean concentration. The vectors are divided into concentration bins based on the colour scale in the legend.**

*10- Figure 12. Panel (a) the numbers of ticks could be harmonized by making it the same (4?).*

Ticks are now harmonized. This figure has also been moved to the SI as, on further consideration, we feel it isn't crucial to the main storyline (now Figure S13).

[Figure]

**Figure S13 – PAH factor mass spectra showing The PAH families: UnSubPAH, MPAH, OPAH, NOPAH and APAH. *The peak at m/z 252 relating to the ion [C₂₀H₁₂]⁺ is a list of PAHs overlapping in mass and includes benzo[b]-, benzo[j]- and benzo[k]flouranthene, along with benzo[a]- and benzo[e]pyrene.**

*11- Figure 14. I recommend increasing the resolution for the top panel (VK diagrams) the dots are lost even in the electronic version if zoomed in.*

Attempts to increase the resolution weren't successful so we increased the marker size instead to show the data more clearly in this figure (now Figure 11).

[Figure]

**Figure 11 – Van Krevelen (VK) diagrams and median diurnal cycles for elemental ratios during each measurement period. Each VK diagram contains the H:C vs O:C data for the PMF solutions and the raw AMS measurements which are coloured based on the time of the measurement. The carbon oxidation states ($\overline{OS}_C \approx 2O/C - H/C$) are shown using grey dashed lines and the functional group gradients are shown using solid grey lines. The blue and red dashed lines demarcate the region where published ambient OOA measurements are commonly found (Ng et al., 2011).**

*12- In the abstract, I recommend adding a mention that sulfate is the largest mass fraction for the pre-monsoon and monsoon periods.*

Additional text has been added to the abstract which now reads as follows:

"…Old Delhi is one of the most polluted locations in the world, and $PM_1$ concentrations reached ~750 µg m$^{-3}$ during the most polluted period, the post-monsoon, where $PM_1$ increased by 178% over the pre-monsoon period. Sulphate contributes the largest inorganic $PM_1$ mass fraction during the pre-monsoon (24%) and monsoon (24%) periods with nitrate contributing most during the post-monsoon (9%). The organics dominate the mass fraction (54-68%) throughout the three periods and using positive matrix factorisation (PMF) to perform source apportionment analysis of organic mass, two burning-related factors were found to contribute the most (35%) to the post-monsoon increase…"

*In the methods section*

*13- at lines 129-131 the Authors mention that they calibrated the AMS "throughout the campaign". I recommend adding a sentence explaining how many times and when (e.g., before, middle, and after?).*

Details on the timings of the calibrations have been added: "The HR-TOF-AMS was calibrated fortnightly over the three campaigns (11 calibrations in total) for its ionisation efficiency of nitrate (IE) and the relative ionisation efficiency (RIE) of other inorganic compounds using nebulised 300 nm ammonium nitrate, sulphate and chloride."

A summary table has also been added containing the calibration results (Table S2).

**Table S2 – Relative ionisation efficiencies (RIE), ionisation efficiencies (IE) and collection efficiencies (CE)**

| Season | IE | RIE $NH_4^+$ | RIE $SO_4^{2-}$ | RIE $Cl^-$ | CE |
|---|---|---|---|---|---|
| Pre-monsoon | 2.92E-07 | 4 | 1.45 | 2.07 | 0.5 |
| Monsoon | 2.92E-07 | 4 | 1.45 | 2.07 | 0.5 |
| Post-monsoon preflux period (11/10/18 - 03/11/18) | 2.89E-07 | 4 | 1.45 | 2.07 | 0.5 |
| Post-monsoon Diwali period (05/11/18 - 14/11/18) | 3.14E-07 | 4 | 1.45 | 1.05 | 0.8 |
| Post-monsoon post-Diwali (14/11/18 - 23/11/18) | 3.14E-07 | 4 | 1.45 | 1.05 | 0.5 |

*14- At lines130 to 135 the Authors mention that in their analysis they had to use different CEs to match the "PM2.5 filter measurements". I recommend expanding this sentence explaining which measurements they are referring to, carried out by which group, with which instrument, and at what time resolution.*

Details have been added to the measurements we refer to: "A collection efficiency (CE) of 0.5 was confirmed through comparisons with gravimetric $PM_{2.5}$ filter measurements taken throughout the pre- and post-monsoon campaigns by Birmingham University. The AMS measurements were added to the BC measurements to give total $PM_1$ which was averaged to match the sample intervals of the filters (6 and 12 h)." More details on this is being published in the companion paper to this study which includes filter measurement comparison graphs (Reyes-Villegas et al., 2020).

*15- At lines 146 -148 the sentence "Therefore, only peaks which significantly improved the open and closed signal residuals were fitted regardless of the residuals in the difference (diff = open – closed) signal." is unclear and leaves the reader wonder which peaks were not included. I understand that the fitting at higher m/z is tricky, but I am wondering if the authors can modify or expand on the sentence to clarify the process to the reader, maybe explaining which peaks were not fitted and why.*

We have expanded the sentence in the text to inform the reader of the fitting procedure and to ensure the explanation is more understandable to the reader: "The PIKA software allows the user to fit peaks based on a reduction in the residuals between the measured signal and the fit. There are measured signals for open (when the chopper is open), closed (when the chopper is closed) and diff (= open – closed), meaning there are also three sets of residuals. Neighbouring peaks may overlap and cause the diff residuals to improve whilst not improving the open and closed residuals. This becomes more relevant when moving to higher *m/z* ions as peaks become broader. Therefore, only peaks which significantly improved the open and closed signal residuals were fitted."

*16- Lines 163-165: ". . . black carbon (BC) measurements which were taken using an Aethalometer AE-31 and corrected for by a Single Particle Soot Photometer (SP-2; Droplet Measurement Technology, Boulder, CO) (Reyes-Villegas et al., 2020).". This sentence is quite vague. I understand that there is a reference to look up, however, I recommend adding a short sentence giving a few more details, e.g., explaining briefly 1) how the Aethalometer data were corrected 2) if/when and for how long the SP2 was co-located with the Aethalometer.*

The correction method is now included along with the number of days the SP2 was co-located with the Aethalometer: "The AMS measurements were compared to a number of co-located instruments including black

carbon (BC) measurements which were taken using an Aethalometer AE-31. The BC measurements were corrected using the Weingartner method (Weingartner et al., 2003) and by using reference measurements from a Single Particle Soot Photometer (SP-2; Droplet Measurement Technology, Boulder, CO) which was co-located for ~7-days (Reyes-Villegas et al., 2020)." More details are also presented in the cited reference (Reyes-Villegas et al., 2020).

*Results*

*17- Lines 235-238: here and in a few other parts, the authors cite "personal communication with Ben Langford". In all those cases I think that this information should be removed as it doesn't seem critical to the point of the sentences unless a paper has been published in the meantime and can be properly referenced.*

We have updated the citation which refers to the abstract of a talk given at the European Aerosol Conference which summarised the results (Di Marco et al., 2019).

18- Line 458: "UnSubPAHs" acronym not defined

A definition has been added: "The PAH composition of COA is mainly unsubstituted PAHs (UnSubPAHs) and also contributes...."

*Conclusions*

*19- Lines 894-895: "These high post-monsoon concentrations have been linked to an increase in burning emissions mainly from crop residue and solid fuel." Are higher concentrations only due to an increase in burning emissions or the boundary layer height affect these concentrations as well? If that's the case, I recommend adding it here.*

We agree and have added this statement within the quoted sentence: "These high post-monsoon concentrations have been linked to an increase in the boundary layer height affect and an increase in burning emissions, mainly from crop residue and solid fuel."

*Supplementary Information*
*20- Figure S2 y-axis label too small*
All figures relating to multilinear regression analysis have been increased in font size.

[Figure]

**Figure S1 – Trilinear regression analysis results for the PMF solutions taken from the all-periods-combined case. Results are shown for the fit using (a) CO, (b) NO$_x$ and (c) BC as external tracers. Gradient contributions for factors SFOA, COA and HOA are shown alongside the background concentration of the tracer (black) which is estimated using the intercept of the linear regression. The chi-square value (red markers), the Q/Q$_{exp}$ (blue markers) and the chosen final solution (labelled with a blue arrow) are also shown below.**

*21- Page 12: "Error! Not a valid bookmark self-reference." Should be "Table S2"*

This has been corrected.

*22- Figure S8: "Polar graphs showing the concentrations . . ." add units of concentrations.*

Units of concentration have been added to this caption: "**Polar graphs showing the concentrations (in μg m$^{-3}$) by wind direction for…**"

*23- Figure S9 "Mean" and "calm" not legible.*

The Figure S9 has been altered to show wind vectors that are sized based on the contribution to the mean. The font size has also been increased.

[Figure]

**Proportion contribution to the mean (%)**

**Figure S9 – Chloride pollution roses for each diurnal hour for all measurement periods combined, where the data is binned into 30° wind vectors and the size of each bin is proportional to its contribution to the mean concentration. The counts are divided into concentration bins based on the colour scale in the legend. Units are µg m⁻³.**

*24- Figure S13: add that the points not labeled neither "Delhi" nor "Chack2018" come from Table S3.*

The caption has been updated to make it clearer that all points included in the Van Krevelen diagram are from Table S6: "**Van Krevelen (VK) diagram for the mass spectra of the organic aerosol factors listed in Table S6. The data labelled with Chak2018 are from the study by Chakraborty et al. (2018) and those labelled with Delhi are values from this study.**" (now Figure S15).

*25- Page 20: "The factor mass profiles and their diurnal cycles during each measurement period are summarized in Figure S14". I think it should be "Figure S15"*

This has been corrected and Figure S15 is now Figure S17: "The factor mass profiles and their diurnal cycles during each measurement period are summarised in Figure S17"

*26- Figure S15 and S16: y-axis labels are too small*

All axis labels have been increased in font size (now Figures S17 and S18).

*Finally, reading the manuscript I have been wondering why the Authors decided not to run the PMF in bootstrap mode for the 7 solutions combined periods.*

We did conduct a bootstrapping analysis. However, the different factors show a high degree of correlation because they share common controls through boundary layer dynamics and temperature. This is a situation in which bootstrapping analysis can often be incorrect or ambiguous (Ulbrich, 2011).

It may be possible to solve this issue using a different resampling method. The PET tool used in this study resamples the data matrix using subsets of original rows (mass spectra) and randomly replaces them with other rows from the original matrix (Ulbrich, 2011). Other software, such as the EPA PMF software, use block bootstrapping which is likely a better resampling method for this dataset. It attempts to account for the effects of serial correlation (or correlation between factors) by resampling via randomly selecting non-overlapping time periods or 'blocks' (i.e. of length 3-days) and creating a new dataset from these blocks (Paatero et al., 2014). This however does not account for large variations between the pre-monsoon and the post-monsoon (e.g. SVBBOA goes from ~0 to ~20 μg m$^{-3}$) because, for example, if a 'block' from the pre-monsoon is placed next to a block from the post-monsoon, this creates a large amount of variation that is not due to model error/solution instability. This has been noted before in previous studies that have implemented other resampling methods to counteract this (Hemann et al., 2009).

In order to account for seasonal variance, we could address this issue by doing bootstrapping on a seasonal basis, but such a detailed analysis is beyond the scope of this study. We would also be required to resample on a short time scale which is computationally demanding for a matrix with $2.39673 \times 10^7$ data points and is currently not supported by software such as EPA PMF. The additional issue is the time series dependence on the boundary layer which is likely not going to be accounted for when using current resampling methods or bootstrapping on a seasonal basis.

However, the variance can be explored by varying the SEED and these results show variance in the solutions. To deal with the problem of potential ambiguity, we applied multilinear regression analysis on multiple SEEDS as a more appropriate method for obtaining a solution in this scenario, which is an established method to deal with such situations (Allan et al., 2010; Young et al., 2015; Elser et al., 2016; Reyes-Villegas et al., 2016), as we were able to compare against multiple external tracers.

Nevertheless, in response to this comment, we have conducted a further bootstrapping analysis (100 iterations) on four selected possible all-combined solutions (6f_ac_S0_C1, 6f_ac_S3_C1, 7f_ac_S1_C1 and 7f_ac_S4_C1) using the PET tool and this gave favourable results in the multilinear regression analysis. We found that the chosen solution (7f_ac_S1_C1) gave a 5% decrease in the relative standard deviation of profiles and 7% decrease in relative standard deviation of the time series when compared to the nearest possible solution (7f_ac_S4_C1). However, it must be noted that these estimates could be incorrect or ambiguous for the reasons mentioned above.

References:

Allan, J. D., Williams, P. I., Morgan, W. T., Martin, C. L., Flynn, M. J., Lee, J., Nemitz, E., Phillips, G. J., Gallagher, M. W. and Coe, H.: Contributions from transport, solid fuel burning and cooking to primary organic aerosols in two UK cities, Atmos. Chem. Phys., 10(2), 647–668, doi:10.5194/acp-10-647-2010, 2010.

Chakraborty, A., Mandariya, A. K., Chakraborti, R., Gupta, T. and Tripathi, S. N.: Realtime chemical characterization of post monsoon organic aerosols in a polluted urban city: Sources, composition, and comparison with other seasons, Environ. Pollut., 232, 310–321, doi:10.1016/j.envpol.2017.09.079, 2018.

Elser, M., Huang, R. J., Wolf, R., Slowik, J. G., Wang, Q., Canonaco, F., Li, G., Bozzetti, C., Daellenbach, K. R.,

Huang, Y., Zhang, R., Li, Z., Cao, J., Baltensperger, U., El-Haddad, I. and André, P.: New insights into PM2.5 chemical composition and sources in two major cities in China during extreme haze events using aerosol mass spectrometry, Atmos. Chem. Phys., 16(5), 3207–3225, doi:10.5194/acp-16-3207-2016, 2016.

Hemann, J. G., Brinkman, G. L., Dutton, S. J., Hannigan, M. P., Milford, J. B. and Miller, S. L.: Assessing positive matrix factorization model fit: A new method to estimate uncertainty and bias in factor contributions at the measurement time scale, Atmos. Chem. Phys., 9(2), 497–513, doi:10.5194/acp-9-497-2009, 2009.

Di Marco, C. D., Langford, B., Cash, J. M., Mullinger, N., Helfter, C. and Nemitz, E.: Source apportionment analysis applied to aerosol eddy-covariance fluxes in Delhi, European Aerosol Conference. [online] Available from: https://www.costcolossal.eu/specialsessioneac2019/, 2019.

Ng, N. L., Canagaratna, M. R., Jimenez, J. L., Chhabra, P. S., Seinfeld, J. H. and Worsnop, D. R.: Changes in organic aerosol composition with aging inferred from aerosol mass spectra, Atmos. Chem. Phys., 11(13), 6465–6474, doi:10.5194/acp-11-6465-2011, 2011.

Paatero, P., Eberly, S., Brown, S. G. and Norris, G. A.: Methods for estimating uncertainty in factor analytic solutions, Atmos. Meas. Tech., 7(3), 781–797, doi:10.5194/amt-7-781-2014, 2014.

Reyes-Villegas, E., Green, D. C., Priestman, M., Canonaco, F., Coe, H., Prévôt, A. S. H. and Allan, J. D.: Organic aerosol source apportionment in London 2013 with ME-2: Exploring the solution space with annual and seasonal analysis, Atmos. Chem. Phys., 16(24), 15545–15559, doi:10.5194/acp-16-15545-2016, 2016.

Reyes-Villegas, E., Panda, U., Darbyshire, E., Cash, J. M., Joshi, R., Langford, B., Di Marco, C. F., Mullinger, N., Acton, W. J. F., Drysdale, W., Nemitz, E., Flynn, M., Voliotis, A., McFiggans, G., Coe, H., Lee, J., Hewitt, C. N., Heal, M. R., Gunthe, S. S., Shivani, Gadi, R., Singh, S., Soni, V. and Allan, J. D.: PM1 composition and source apportionment at two sites in Delhi, India across multiple seasons, Atmos. Chem. Phys. Discuss., 2020, 1–19, doi:10.5194/acp-2020-894, 2020.

Ulbrich, I. M.: Characterization of Positive Matrix Factorization Methods and Their Application to Ambient Aerosol Mass Spectra, Chem. Biochem. Grad. Theses Diss., Paper 35 [online] Available from: https://core.ac.uk/download/pdf/54848806.pdf, 2011.

Weingartner, E., Saathoff, H., Schnaiter, M., Streit, N., Bitnar, B. and Baltensperger, U.: Absorption of light by soot particles: determination of the absorption coefficient by means of aethalometers, J. Aerosol Sci., 34(10), 1445–1463, doi:10.1016/S0021-8502(03)00359-8, 2003.

Young, D. E., Allan, J. D., Williams, P. I., Green, D. C., Harrison, R. M., Yin, J., Flynn, M. J., Gallagher, M. W. and Coe, H.: Investigating a two-component model of solid fuel organic aerosol in London: Processes, PM1 contributions, and seasonality, Atmos. Chem. Phys., 15(5), 2429–2443, doi:10.5194/acp-15-2429-2015, 2015.